# Accelerated Aging Characterizes the Early Stage of Alzheimer’s Disease

**DOI:** 10.3390/cells11020238

**Published:** 2022-01-11

**Authors:** Alessandro Leparulo, Marta Bisio, Nelly Redolfi, Tullio Pozzan, Stefano Vassanelli, Cristina Fasolato

**Affiliations:** 1Department of Biomedical Sciences, University of Padua, Via U. Bassi 58/B, 35131 Padua, Italy; alessandro.leparulo@unipd.it (A.L.); marta.bisio@unipd.it (M.B.); nelly.redolfi@unipd.it (N.R.); tullio.pozzan@unipd.it (T.P.); 2Neuroscience Institute-Italian National Research Council (CNR), Via U. Bassi 58/B, 35131 Padua, Italy; 3Venetian Institute of Molecular Medicine (VIMM), Via G. Orus 2B, 35129 Padua, Italy; 4Padua Neuroscience Center (PNC), University of Padua, Via G. Orus 2B, 35129 Padua, Italy

**Keywords:** Alzheimer’s disease, PS2APP, presenilin-2, amyloid-β, slow oscillations, delta waves, functional connectivity, phase-amplitude-coupling, UP-DOWN states, spikes

## Abstract

For Alzheimer’s disease (AD), aging is the main risk factor, but whether cognitive impairments due to aging resemble early AD deficits is not yet defined. When working with mouse models of AD, the situation is just as complicated, because only a few studies track the progression of the disease at different ages, and most ignore how the aging process affects control mice. In this work, we addressed this problem by comparing the aging process of PS2APP (AD) and wild-type (WT) mice at the level of spontaneous brain electrical activity under anesthesia. Using local field potential recordings, obtained with a linear probe that traverses the posterior parietal cortex and the entire hippocampus, we analyzed how multiple electrical parameters are modified by aging in AD and WT mice. With this approach, we highlighted AD specific features that appear in young AD mice prior to plaque deposition or that are delayed at 12 and 16 months of age. Furthermore, we identified aging characteristics present in WT mice but also occurring prematurely in young AD mice. In short, we found that reduction in the relative power of slow oscillations (SO) and Low/High power imbalance are linked to an AD phenotype at its onset. The loss of SO connectivity and cortico-hippocampal coupling between SO and higher frequencies as well as the increase in UP-state and burst durations are found in young AD and old WT mice. We show evidence that the aging process is accelerated by the mutant PS2 itself and discuss such changes in relation to amyloidosis and gliosis.

## 1. Introduction

Alzheimer’s Disease (AD) is a worldwide plague affecting millions of people in aging societies. It has huge social and economic costs, also considering the lack of effective therapies; its prevalence is expected to increase, especially in low- and middle-income countries, with an estimated 139 million affected individuals by 2050 (Dementia facts & Figures. Available online: https://www.alzint.org/about/dementia-facts-figures/ accessed on 21 December 2021). Deterioration of the cognitive function is thus considered one of the greatest health threats of old age, but what distinguishes cognitive decline linked to “normal” aging from the first signs of dementia is still the subject of debate [1]. Very little is known about the processes that cause conversion from healthy aging to mild cognitive impairment (MCI) and, from this condition, to AD, even if improvements in electroencephalographic (EEG) recordings potentially allow to follow MCI to AD conversion [2,3]. Thus, there is an urgent need for biomarkers that allow identification of AD staging and even more for markers associated with a high probability of disease worsening [4,5].

Cognitive impairments are closely linked to brain circuitry dysfunctions, and the latter are firstly associated with alterations of neuronal connectivity rather than with cell death [6,7,8,9,10]. The aging process itself has been linked to synaptic dysfunctions and the molecular mechanisms involved are starting to be identified [11,12,13]. Besides neuronal deficits, also alterations occurring in astrocytes, glial cells and factors of the innate immunity have been implicated directly in a process known as “inflammaging” [14,15,16,17].

Memory decline, associated with accumulation of amyloid-beta (Aβ), its related peptides and neurofibrillary tangles (NFTs) as well as to neuroinflammation, has thoroughly been investigated in both AD patients and different mouse models of the disease [18,19]. From the functional point of view, there is now a general consensus that AD is characterized by an excitation/inhibition imbalance at different circuitry levels that starts before plaque seeding [20], with neuronal hyperexcitability being consistently found upon single cell and extracellular local field potential (LFP) recordings, as well as with EEG or electrocorticographic (ECG) analyses [21,22,23,24]. Both in vivo studies, with anesthetized or awake subjects [25,26,27,28], and in vitro analyses, with cortical slices or isolated hippocampi, have established defective gamma (30–90 Hz) oscillations [28,29,30] and theta-gamma cross-frequency coupling (CFC) [31], a phenomenon playing a primary role in memory encoding and retrieval, both in humans [32,33,34,35,36] and mice [24,37]. The concept that, in AD, cognitive abnormalities are causally linked to network hyperexcitability and altered oscillatory rhythms, is also supported by the fact that manipulation of network activities, either artificially or following sensory stimulation, can rescue brain functionality and behavior in rodent models [30,38,39] and, possibly, also in humans [4,40,41].

We have recently shown that, in pre-depositing AD mice, brain spontaneous activity is characterized by marked reduction in the relative power of slow oscillations (SO, 0.1–1.7 Hz) [42]. In addition, coupling of SO to higher frequencies as well as SO connectivity are impaired at the hippocampal and cortical levels in plaque-seeding mice [42]. As reported by other groups [38,43,44,45], defective oscillatory activity in the very low frequency range (<3 Hz) is a prominent feature of AD, at its early stages. We have here expanded the study of spontaneous brain activity to aged AD mice, also taking into account alterations in UP/DOWN-states and spiking activity. By this integrated approach we have analyzed the cross-talk between aging and AD progression in terms of basal network activity, and defined which specific changes are linked to AD progression but are distinct from the aging process. Furthermore, we show evidence that young AD mice have striking similarities to old control mice.

## 2. Materials and Methods

### 2.1. Animals

The homozygous transgenic (tg) mouse lines B6.152H (AD) and PS2.30H were kindly donated by L. Ozmen (F. Hoffmann-La Roche Ltd., Basel, Switzerland) [46]. In these lines, human *APP* and *PSEN2* transgenes are driven by mouse Thy-1.2 and prion protein promoters. The B6.152H were generated in a C57BL/6J background. The PS2.30H were originally backcrossed to C57BL/6J mice for seven or more generations (>95% C57BL/6J background). As a control, we used C57BL/6J (WT) mice. All the animals were reared in an SPF animal facility, in 12/12-h light/dark cycles, with free access to food and water. For each genotype, mice were from 3-, 12- and 16-month-old cohorts, with either two-, three, and four-week-tolerance, respectively. Only female mice were used because in B6.152H, amyloidosis occur earlier with respect to males [46]. Estrus cycle was not check because the mice which were studied in multiple sessions for each genotype/age group, derived from 3 to 6 different litters, possibly preventing bias by randomizing the estrus cycle. Indeed, no correlation was found between estrus cycle and EEG/LFP recordings in different brain regions of AD mice [47]. All experimental procedures were performed according to the European Committee guidelines (decree 2010/63/CEE) and the Animal Welfare Act (7 USC 2131), in compliance with the ARRIVE guidelines, and were approved by the Animal Care Committee of the University of Padua and the Italian Ministry of Health (authorization decree 522/2018-PR).

See Appendix A for animal preparation and surgery, signal acquisition, electrophysiological data processing and analyses, immunohistochemistry.

### 2.2. Data Presentation and Statistical Analyses

Throughout the text, the data were expressed as mean ± SEM. Unless otherwise indicated, the figures show the data as boxplots indicating the median (dashed line), the mean (cross), the 25th and 75th percentiles of the distributions (edges of the box) and the upper and lower extremes (whiskers); outliers are not shown. The non-parametric Kruskal–Wallis test was used for statistical analysis between ages in all cohorts (3, 6, 12, and 16 months). The non-parametric Wilcoxon rank sum test was used for pairwise comparison within ages to study the same parameters between the control group (3-month-old WT mice) and each other group, separately for age and genotype (* *p* < 0.05; ** *p* < 0.01; *** *p* < 0.001). Whenever indicated, comparison was also carried out with respect to 3-month-old AD mice (^o^ *p* < 0.05; ^oo^ *p* < 0.01). Unless otherwise specified, all the reported analyses were based on the following numbers of mice: 13, 10, 7 for WT, and 16, 10, 7 for AD at 3-, 12- and 16-months of age, respectively; within each group, mice derived from at least three different litters; the 3-month-cohorts include data from previous work [42]. For analysis of UP/DOWN-states and spiking activity, we included the recordings from the 6-month-old cohorts of our previous work [42].

## 3. Results

### 3.1. Total Power Is Reduced in Young AD and Old WT Mice

To establish how brain network dysfunctions evolve in AD mice, we recorded spontaneous brain electrical activity from anesthetized female mice at 3-, 12-, and 16-months of age, also classified as young, middle-aged, and old mice. We employed the B6.152H mouse line, a double PS2APP transgenic line [46,48], here simply called AD, and wild-type (WT) C57Bl/6J mice, the background strain. LFP recordings were obtained with a multi-site linear probe sampling from the posterior parietal cortex (PPC), through the hippocampal formation (HPF) up to the dentate gyrus (DG) as previously described [42] and summarized in Appendix A. The total power profiles, obtained with the 24 electrodes, look quite different in WT (Figure 1A) and AD (Figure 1B) mice.

Compared to 3-month-old WT mice, young AD mice show a significant reduction in total power, especially at the hippocampal level, being halved in CA1 at the stratum radiatum-lacunosum molecolare (*sr-lm*,1500 µm), a reduction that is maintained in middle-aged AD mice (Figure 1C). The power profile is markedly reduced at all depths recorded in old mice of both genotypes (Figure 1A,B), reaching a reduction in CA1 of approximately 80% (Figure 1C and Appendix A for representative traces of LPF recordings in in the hippocampus of 12- and 16-month-old WT mice). Of note, in young AD mice, the power loss occurs in the absence of plaques and gliosis (Figure 2A–D, Appendix A), which start around 6 months of age [23,46], and are abundantly present in middle-aged (Figure 2E–H) and old AD mice (Figure 2I–L, see also Appendix A).

### 3.2. Power Imbalance in the Low Frequency Range Characterizes Young AD Mice

Total power reduction can occur with or without major changes in the relative contribution of different frequency ranges (Appendix A). We have previously identified a strong power imbalance in the Low frequency range (0.1–4.7 Hz), which selectively occurs in CA1 (*sr-lm*, 1500 µm) of young, pre-depositing AD mice and persists in 6-month-old AD mice, at the beginning of plaque seeding [42]. This power imbalance is mainly due to a significant loss of SO (0.1–1.7 Hz) relative power (−33 ± 8%, *p* = 0.0031) as shown in Figure 3A. To test whether this imbalance is maintained, or even worsened with further plaque accumulation and gliosis, we compared the relative power of the different frequency bands in middle-aged and old AD as well as in WT mice versus young WT mice. With this approach, we were also able to study how spontaneous brain activity changes across ages in WT mice, identifying similarities and differences with the AD phenotype. First, we noticed that, in middle-aged WT mice, in CA1, the SO relative power is stable (Figure 3A), and strikingly similar to that of age-matched AD mice, thus showing a significant recovery compared to young AD mice (+42 ± 10%, *p* = 0.0106). However, in old AD mice, the relative power of SO decreases again (Figure 3A). In contrast, the relative power of delta waves (1.7–4.7 Hz) is quite stable in both genotypes, becoming significantly higher only in old AD mice (+32 ± 13%, *p* = 0.0394) (Figure 3B). The power imbalance in the Low frequency range is also quantified by the SO/delta power ratio (Figure 3C): the strong reduction (−45 ± 8%, *p* = 0.0070) observed at 3, as well as at 6 months of age (−48 ± 8%, *p* = 0.0091, data not shown but see [42]), is absent at 12 months, when plaques and gliosis are abundantly present in hippocampus and cortical areas (Figure 2, Appendix A and [46]). Of note, in old AD mice, the SO/delta power ratio is significantly decreased with respect to young WT mice (−47 ± 14%, *p* = 0.0265, Figure 3C).

Normalized Power Spectral Density (PSD) plots suggest an increase in the contribution of the higher frequency bands in AD mice (Appendix A). In young and old (but not middle-aged) AD mice, there is a significant increase in the theta (4.7–10 Hz) relative power, but no significant change is found in WT mice across ages (Appendix A). The beta (10–25 Hz) band increases in 16-month-old AD mice, and the Fast Gamma (FG, 45–90 Hz) band weighs more in 12- and 16-month-old AD mice (Appendix A). The epsilon (90–190 Hz) band, which contributes marginally to total power, increases in AD mice at all ages, being significantly larger (3–4 times) in middle-aged and old AD mice (Appendix A). Interestingly, middle-aged and old WT mice also show a significant increase in the FG and epsilon bands, respectively (Appendix A), see also Appendix A. Overall, in AD mice in CA1, the power ratio between the Low (0.1–4.7 Hz) and High (4.7–190 Hz) frequencies follows the same trend as the SO/delta power ratio (Figure 3D): (i) the significant reduction observed in young AD mice is recovered in middle-aged AD mice; (ii) but reappears in old AD mice (*p* = 0.0470); (iii) finally, no significant change emerges in WT mice across ages.

### 3.3. Impaired Coupling of SO to Higher Frequencies Is Shared between Young AD and Aged WT Mice

At rest as well as during activities, oscillations in the low-frequency range link distant brain areas and allow for encoding and retrieval of memory through coordination with local high frequency activity [45,49,50,51,52]. At the circuit level, these processes also rely on nesting between low- and high-frequency oscillations from different regions, a phenomenon specifically measured by cross-frequency coupling (CFC) [31]. Disruption of this coupling has been linked to poor task performance and cognitive decline in humans and mice [33,34,35,36,37]. Defective Phase-Amplitude Coupling (PAC), especially between theta and gamma frequencies, has been reported in different AD models [23,24,53,54]. Here, we quantified PAC by computing the General Linear Model (GLM) index for SO and delta waves with respect to higher frequencies, as described in Appendix A. As found in 6-month-old AD mice [42], middle-aged AD mice also show a significant loss of PAC between SO in L4/5 and FG in CA1 (−65 ± 5%, *p* = 0.0079). This loss is not only maintained in old AD mice (−66 ± 11%, *p* = 0.0031), but it is also observed in old WT mice (−50 ± 7%, *p* = 0.0027). Overall, these findings indicate that SO-PAC between PPC and CA1 is significantly reduced in old mice, regardless of genotype, but, in AD mice, it appears at 6 months [42]. We also measured similar reductions in PAC in the reverse (bottom-up) direction, i.e., between SO in hippocampus and FG in PPC, which however start from 12 months in AD mice (Figure 4A). Regarding delta wave PAC with FG, only bottom-up PAC is affected in aged AD and old WT mice (Figure 4B). For delta waves, the GLM indices are lower than those for SO (see Figure 4A,B), consistent with the fact that SO are more suitable to support global activity than delta waves [55]. Furthermore, even the SO—but not delta—PAC with the highest frequency range, the epsilon band (90–190 Hz), was significantly compromised already at 6 months (Appendix A), therefore the PAC based on SO, especially that linked to cortical SO and hippocampal FG, is a sensitive marker of brain aging linked to AD dysfunctions.

### 3.4. Defective SO Connectivity Anticipates the Aging Process in Young AD Mice

The propagation of electrical signals in the low frequency range allows to connect specific brain networks and adjust their activity according to different brain states and tasks. An index of functional connectivity between brain regions is derived from cross-correlation analysis, as previously described [56], and summarized in Appendix A. The changes in SO connectivity displayed in Figure 5 report maximal cross-correlation coefficients of instantaneous amplitudes (upper-right matrices) and corresponding latencies (lower-left matrices) of WT (Figure 5A–C) and AD (Figure 5D–F) mice across ages. In young AD mice we observed a significant reduction of maximal cross-correlation coefficients within the HPF, and between the latter and the PPC (Figure 5A,D, upper-right, black lines). These changes are accompanied by an increase in latencies in CA1 and between deep cortical layers and DG (Figure 5A,D, lower-left, black lines). Altogether these findings suggest a substantial impairment of SO connectivity. Curiously, we observed similar changes in cross-correlation coefficients and latencies also in 12- and 16-month-old WT mice (Figure 5A–C, lower-left, black lines).

To obtain the quantification at the regional level, we averaged the cross-correlation coefficients and latencies from adjacent channels that correspond to defined hippocampal and cortical regions. According to the scheme illustrated in Figure 6A, we identified the regions with statistical significance with respect to young WT mice (Figure 6B) and quantified the respective changes (Appendix A). Regarding similarities to the aging process, a consistent reduction in cross-correlation of SO is primarily found within the DG in AD mice at all ages as well as in middle-aged and old WT mice (Figure 6B and Appendix A). Second, in young AD mice, the SO in L4/6 and L2/3 are delayed with respect to the DG by 57 ± 13 ms (*p* = 0.0061) and 42 ± 10 ms (*p* = 0.0168), respectively (Figure 6B and Appendix A). Similar delays are also found in 12- and 16-month-old mice of both genotypes (Figure 6B and Appendix A).

We can argue that young AD mice partially mimic middle-aged and old WT mice: cross-correlation is reduced in DG and latency is increased in CA1 and between PPC cortical layers and DG. However, young AD mice show a stronger phenotype with further loss of cross-correlation in CA1 and between this region and DG (Figure 6B, Appendix A). It is worth noting that, in middle-aged mice of both genotypes, SO latency is reduced within the PPC layers (Figure 5B,E lower-left, black lines), reaching significance in WT mice (Figure 6B and Appendix A). In the same regions, the cross-correlation is significantly increased in old WT mice (Figure 5C, F upper-right, black lines, Figure 6B and Appendix A), highlighting, at the cortical level, that the presence of compensatory mechanisms are not fully effective in old AD mice.

When the same approach is used to evaluate the functional connectivity supported by delta waves (Appendix A), it is noted that, compared to young WT mice, cross-correlation significantly increases in L4/6 of middle-aged AD and WT mice (Figure 6C,D). In old mice of both genotypes, cross-correlation also increases between these layers and the hippocampus, as well as within the PPC (Figure 6D). Overall, changes in delta connectivity appear similar in AD and WT mice at all ages, perhaps reflecting “healthy” aging, while changes in SO connectivity suggest “pathological” premature aging.

It is worth noting that the age-matched comparison of the SO (or delta) connectivity was much less informative than the analysis across ages: in fact, the comparison of the 12- and 16-month-old AD cohorts with age-matched WT mice results in very few differences from those observed in young mice (Appendix A), probably due to two converging phenomena, the decline in WT mice with age and the slow progression of amyloidosis in AD mice after the abrupt changes that occur in the early stages.

### 3.5. Imbalance in UP- and DOWN-States Marks AD Progression

The ongoing neuronal activity present in the background is reflected by the subthreshold neuronal membrane potential which spontaneously fluctuates between a hyperpolarized DOWN-state and an intermittent, depolarized UP-state [57], from which action potentials arise [57,58,59]. These fluctuations critically determine the firing patterns and functional properties of neuronal circuits, as indicated by both in vivo studies [60,61,62,63] and in vitro and ex vivo approaches [64,65,66]. In this study, we also analyzed isolated spikes and spike bursts (see Appendix A), the latter being critical for synaptic plasticity, information processing and memory encoding [67] as well as for Aβ accumulation [68]. Slow oscillatory activity, as part of LFP signals, is associated with UP- and DOWN-state transitions; therefore, we analyzed the signal recorded in the 0.3–3 kHz frequency range, and compared it with the LFP traces for alterations in spiking activity and in UP- and DOWN-states, as described in Appendix A and in Appendix A. In this type of analysis, we also included the 6-month-cohorts from our previous study [42].

In young AD mice, there is a marked increase in UP-state duration, which lasts approximately 2–3 times longer than in WT mice (Figure 7), reaching significance at all investigated brain levels (Figure 8A and Figure 9). For instance, the UP-state duration goes from 0.75 ± 0.06 s to 3.16 ± 1.49 s (*p* = 0.0272) in CA1 (1500 µm) of young WT and AD mice, respectively. It should be noted that the number of UP-states is slightly but significantly reduced at several levels, being for instance 30% lesser in CA1 (Figure 8B and Figure 9).

In 6-month-old AD mice, the mean firing rate (MFR) increases by approximately 50%, reaching significance at almost all levels (Figure 8). In the deep cortical layers and CA1, there is a doubling of spike number per UP-state, being 19.8 ± 3.8 and 37.9 ± 8.7 (*p* = 0.0530) at CA1 (*sr-lm*, 1500 µm) of 6-month-old WT and AD mice, respectively. At the same level, burst duration similarly increases from 0.31 ± 0.09 to 0.75 ± 0.17 s (*p* = 0.0356) (Figure 8 and Figure 9D). Old AD mice also show significant changes, especially in UP-state duration and number (Figure 9A,B). When considering the same parameters in WT mice of all ages, significant changes are present, but limited to selected regions: for instance, the UP-state duration increases in DG, at 6-months, and in CA1 (1500 µm), at 12 and 16 months (Figure 8 and Figure 9A); moreover, the burst duration increases in CA1 of WT mice by 6 and 16 months, showing a similar trend to that reported in AD mice (Figure 9D), with statistical significance only in DG and cortical layers.

We were also interested in establishing whether these alterations in UP-DOWN states and spiking activity are caused by mutated PS2, Aβ itself, or both. We took advantage of PS2.30H mice that express only the human PS2 mutation N141I linked to FAD [48]. Indeed, neurons of PS2.30H mice, both in brain slices and in primary cultures, show both Ca^2+^ dysregulation [69] and autophagic defects [70], yet in the absence of Aβ42 accumulation up to 12 months [23,69]. We thus analyzed the recordings obtained from the 3- and 6-month-old cohorts employed in our previous work [42], as well as a new cohort of middle-aged PS2.30H. Compared to young WT mice, the increase in UP-state duration and the decrease in UP-state number that characterize young AD mice are also found in young PS2.30H mice, especially at cortical levels (Figure 8), suggesting a primary role played by mutant PS2, in the absence of Aβ accumulation and gliosis [23]. In addition, MFR, spike number and burst duration were significantly increased in young PS2.30H mice, similarly to what observed in 6-month-old AD mice and old WT mice (Figure 8). Altogether these findings confirm a hyperexcitable phenotype that mainly affects young and old mice expressing the mutant PS2 as well as old WT mice, to a lesser extent. Indeed, 30% of FAD patients carrying PS2 mutations show seizures [71].

Interestingly, DOWN-state duration is significantly reduced only in old AD mice at all hippocampal and cortical levels (Figure 8 and Figure 9C). For instance, in CA1 (*sr-lm*, 1500 µm), the DOWN-state duration changes from 1.03 ± 0.30 to 0.64 ± 0.04 s (*p* = 0.0031), a phenomenon that occurs in the presence of a longer UP-state duration, thus worsening the imbalance in UP-DOWN cycles.

### 3.6. Amyloidosis and Inflammation in AD and Aged WT Mice

In the double transgenic AD mouse line B6.152H, plaque seeding starts at 5–6 months of age [23,46]. In these mice, intraneuronal Aβ (iAβ) is detectable from one month of age, in the cortex, subiculum and CA1 with two different antibodies: 4G8 which recognizes Aβ, full-length APP and βC-terminal-fragments (βCTFs) (Appendix A), as well as McSA1 which selectively targets Aβ peptides [72,73] (Appendix A). Signs of dystrophic axons, as detected by Lamp1 aggregates, in proximity to plaques and activated microglia (Iba1) [74,75], are clearly present in 12- and 16-month-old AD mice (Appendix A). These types of alterations are not detectable in WT mice up to 16 months, except for a tendency for increased activation of astrocyte and microglia with age, as detected by GFAP and Iba1 reactivity, respectively, but at a level much lower than that found in AD mice (Figure 2K,L; Appendix A). In particular, in aged WT mice, microglia activation has been documented by µPET and immunohistochemistry [76]. In adult WT mice, we occasionally found iAβ, as defined by 4G8 reactivity (Figure 2G,H; Appendix A), yet it was not reproduced consistently with MsCA1 (Appendix A), likely because 4G8 also detects APP/βCTFs, thus reflecting a variability in the level of APP expression in mice [46,77].

## 4. Discussion

Using a multisite linear probe that traverses the PPC and reaches the dorsal hippocampus, we have previously identified profound alterations in the spontaneous oscillatory activity of 3- and 6-month-old AD mice under anesthesia [42]. In this work, we extended this type of analysis to middle-aged and old AD and WT mice, looking for electrical signs that mark the progression of amyloidosis and gliosis but are distinct from the aging process.

### 4.1. Specific Markers of Brain Changes in Young AD Mice

Following a comparison with 3-month-old WT mice, we discovered that the Low/High power imbalance that occurs mainly in CA1 of young AD mice, is a feature of early AD, as it is absent in middle-aged and old WT mice. We have confirmed that the reduction of the Low/High power ratio has two causes: firstly, a dramatic loss of SO compared to delta waves and, secondly, a significant increase in High frequencies, at the level of the theta and beta bands. Interestingly, the SO contribution is recovered in middle-aged AD mice, being close to the level of WT mice, but decreases again in old AD mice. A Low/High power imbalance and especially a reduced SO/delta ratio can therefore be considered early biomarkers of prodromal AD. Interestingly, neural activity driven by SO and delta waves have competing roles in sleep-dependent memory consolidation, with SO promoting memory reinforcement and delta waves supporting forgetting [55].

### 4.2. Premature Aging Also Characterizes the Brain Alterations in AD Mice

Surprisingly, some alterations found in 3- and 6-month-old AD mice are also present in middle-aged and old WT mice. In particular, the loss of SO connectivity within the hippocampus, defined by reduced cross-correlation and increased latency, as well as the loss of coupling between SO and FG, to and from PPC and CA1, are found in middle-aged and old WT mice, respectively. The presence of both types of alterations in young AD mice can thus be interpreted as a sign of “premature aging”. Furthermore, in humans, the presence of defective resting-state functional connectivity (rs-fMRI) characterizes cerebral aging and accelerated aging appear to be a feature of FAD in the pre-clinical phases [78].

There are three other features present in young AD mice that are also found in aged WT mice. The first is a marked reduction in total power, especially at the hippocampal level; the second is an increase in the contribution of the higher frequencies, FG and epsilon bands, in middle-aged and old WT mice, respectively; the third is the increase of the MFR and of the duration of UP-states and bursts. In contrast, the increase in delta connectivity within the PPC and between this latter and the HPF occurs similarly in middle-aged and old WT and AD mice, possibly highlighting a pure aging feature.

The fact that, with aging, WT mice display partly similar brain oscillatory patterns to those of AD mice, allows us to draw two conclusions: (i) First, AD and WT mice share similar aging processes, but AD mice show accelerated aging; (ii) secondly, middle-aged and old WT mice show brain alterations that partially mimics the AD phenotype. It has been reported that “normal” aging is commonly associated with progressive impairments in hippocampal-dependent memory [12]. As far as the murine models are concerned, if kept inbred, the C57Bl/6J strain can be employed as a model of mild cognitive impairment (MCI); with 16-month-old C57Bl/6J mice showing both histological and cognitive changes reminiscent of the early stages of prodromal AD [11,17,79]. On purpose, both WT and AD mice were kept outbred in our work. Consistently, after histopathological inspection, the elderly WT mice employed in this study show no signs of amyloidosis; furthermore, in these mice up to 16 months of age, the detection of iAβ is questionable: it appears to be present in some aged mice, with a pattern partly similar to that observed in young AD mice, but only with the less specific antibody (4G8) that also detects APP/βCTFs. On the contrary, compared to young mice, a trend towards higher levels of gliosis is evident in old WT mice, marked by the reactivity to GFAP and Iba1, confirming previous findings [74,76] and consistent with “inflammaging”.

Curiously, we also found some alterations in SO connectivity that are present only in aged WT mice, perhaps as an expression of a resilient phenotype typical of “healthy” aging: middle-aged and old WT mice show greater SO connectivity between the superficial and deeper layers of the PPC, in terms of reduced latency and increased cross-correlation, respectively, a feature missing in aged AD mice.

### 4.3. AD Mice Show More Brain Changes

Despite the similarities with the aging process, the alterations in functional connectivity appear more pronounced in AD mice, highlighting specific effects related to the progression of amyloidosis and/or gliosis. Indeed, young and middle-aged AD mice have a further reduction of SO cross-correlation within CA1 and between the latter and DG.

In humans as well as in mouse models, AD is characterized by hyperexcitability at both hippocampal and cortical levels [6,7,10], measured by in vivo approaches, either in awake or anesthetized/sleeping subjects [22,25,26,27,28], as well as by various in situ assays [24,28]. In this work, the analysis of the UP-DOWN states indicates that AD mice have a marked increase in the UP-state duration, at all brain levels studied, with the UP-state number also being reduced. Significant alterations in UP-state duration were also found in young mice of the PS2.30H line—carrying only the PS2 mutation N141I, linked to FAD [23,48]—a highly informative result on the possible underlying mechanisms (see below). The high spiking activity, both in terms of increased MFR and duration of the bursts, characterizes both young AD mice and old WT mice. Therefore, hyperexcitability seems to be a characteristic of the early AD stages [23], probably linked to accelerated aging, as it is also largely independent of marked Aβ accumulation and/or seeding. Furthermore, old AD mice are further affected by a shorter duration of the DOWN-state, which aggravates the AD phenotype.

### 4.4. Mechanistic Insights into Brain Network Alterations

From a mechanistic point of view, the fact that cortico-hippocampal PAC, involving SO and FG, is similarly damped in young AD and old WT mice, clearly indicates that it cannot be ascribed to plaque seeding and/or gliosis. Furthermore, we have previously shown that the same type of alteration is also found in young mice of the single transgenic line PS2.30H, but not in *PSEN2*^-/-^ mice [42], which also lack the histological features of AD. The new data, obtained in aged WT mice, support the hypothesis that these alterations, similar to those induced by the expression of mutant PS2, are directly involved in a “pathological” aging process. In fact, it has been demonstrated that, in neurons from AD patients, early AD is accompanied by an increase in the expression level of *PSEN2* mRNA and PS2 protein through the loss of the transcriptional regulator REST [80]. Likewise, C57Bl/6J mice also show increased PS2 levels with aging [81]. We have accumulated evidence that both endogenous and FAD-linked mutant PS2 are directly involved in Ca^2+^ homeostasis [82]. PS2 is a therefore a good candidate to explain the impaired functional connectivity that strongly depends on neuronal Ca^2+^ signaling [38,43] and the propagation of Ca^2+^ waves through the glial network [83]. It is worth noting that, in these mouse lines, the mutant PS2 is expressed under the prion-protein promoter, thus exerting relevant effects also at the glial level. We have shown here that, in PS2-expressing mice, the UP-state number is decreased against a marked increase in the UP-state duration, a finding suggestive of a direct role of astrocytes, given their capability to control the UP-states of surrounding neurons [84,85]. According to these authors, it is the Ca^2+^ activity of astrocytes that strictly determines the DOWN- to UP-state transition both in vitro and in vivo. Consistently, FAD-linked PS2 mutations reduce Ca^2+^ transients in neurons and astrocytes [69,82,86,87], and increase excitability both in vitro [69] and in vivo [23,42], possibly explaining the failure of the DOWN to UP transition.

As for the specific molecular mechanisms and cellular pathways responsible for increasing the UP-state duration, multiple explanations can be foreseen. Both short-term synaptic depression and activity dependent K^+^ conductance have been implicated in the dysfacilitation of the UP-state [88,89]. The latter is considered more relevant in the termination of the UP-state while the former better controls the synchrony of these transitions. Activity-dependent K^+^ conductance is controlled by multiple factors, among which intracellular ATP level and Ca^2+^ transients play major roles. It should be noted that, in addition to the reduction of Ca^2+^ transients, PS2 mutations linked to FAD-also reduce ATP levels, compromising the functionality of mitochondria as well as the redox state [82,90,91], while toxic Aβ oligomers can cause further metabolic and Ca^2+^ dysregulation [92,93]. During slow-wave-sleep, synchronous transitions to DOWN-states are also actively controlled by thalamic sensory inputs via the activation of inhibitory interneurons acting on metabotropic GABA-B receptors [94], a network known to be altered in AD [95,96]. An important role in UP- and DOWN-state transition is also played by the inward rectifying K^+^ channels (GIRK) dependent on G proteins, in particular those linked to GABA-B receptors at the hippocampal level, as shown in mouse AD models [97]. Further studies are needed to address specific synaptic and neuro-modulatory dysfunctions, linked to altered PS2 expression.

From the histopathological point of view, iAβ was not consistently detected in aged WT mice. Our previous data also indicate that, in middle-aged WT mice, Aβ_42_ levels are at least 100 times lower than those found in young AD mice [23], making a direct role of amyloidosis in elderly WT mice unlikely. Similarly, in these mice, we found no signs of microglia activation, of the type found in aged AD mice, namely the Lamp1/Iba1/Aβ co-staining, which marks the dystrophic neurons around plaques, suggesting that in glial cells much more subtle changes take place during the aging process. Concerning neuroinflammation, a specific population of disease-associated astrocytes has been identified in 5xFAD mice, middle-aged WT mice and aging human brains [98]. Whether these types of astrocytes are also responsible for the accelerated aging, described here in PS2APP mice, deserves further investigation, given the complexity of the field as recently highlighted [99].

### 4.5. Study Relevance

Hyperexcitability is an important feature of AD in both humans and mouse models [7,26]. In this work we show that the spontaneous brain activity of AD mice is characterized by a marked increase in duration of the UP-state and burst activity, that occurs as early as 3 months, mainly in CA1. This type of hyperexcitability increases at 6 months, further invading all cortical layers, and persists in old AD mice, and it is also present in old WT mice, albeit to a lesser extent. These data are consistent with early hyperexcitability, in the form of silent seizure activity, detected in humans at the hippocampal level, in the absence of cortical alterations [26] as well as in different AD mouse models [10].

Concerning the power imbalances, it is worth noting that, in humans, an increase in theta relative power was detected by the quantitative electroencephalogram (QEEG) in normal elderly subjects, subsequently identified as “Decliners” to dementia with respect to “Non-Decliners” [5]. In our study, young AD mice also show a significant increase in theta relative power. In most EEG studies, SO and delta waves are grouped together, these studies being mainly focused on theta-gamma frequencies due to their recognized role in brain disease and cognitive decline [7]. The relevance of SO in AD is only now emerging, especially considering the role this frequency range plays on memory consolidation during rest and sleep [45]. By using EEG recordings in sleeping subjects, it was demonstrated that the loss of slow wave activity (<1 Hz), but not delta waves (1–4 Hz) predicts levels of Aβ aggregation as measured by PET [52].

In PS2APP mice, the loss of SO relative power is a specific feature of early AD, while the increase in the higher frequency bands, especially FG and epsilon, is shared by both AD and aged WT mice, highlighting features of cognitive decline associated with the aging process. The PS2APP line does not show tau pathology and massive neurodegeneration, being better considered a model of early amyloidosis, that mimics the initial stages of the disease [76]. Curiously, in these mice, the early Low/High power imbalance is recovered at 12 months of age, and the alterations in functional connectivity are also less marked than at both 3 and 16 months of age. These findings are consistent with the attenuation of the behavioral phenotype in the delayed-matched to position performance, which was observed only at 12 months in PS2APP mice [100]. Overall, these findings suggest compensatory mechanisms at work in middle-aged PS2APP mice that prevent linear disease progression, possibly making this mouse line also a good model of late-onset AD.

### 4.6. Study Limitations

We noted that, using age-matched comparisons, middle-aged and old AD mice show very little differences from young AD mice. We interpreted this finding as an indication of converging aging processes. Therefore, for this study we only performed pairwise comparisons of each age/genotype cohort with 3-month-old WT mice to highlight significant differences, avoiding correction for multiple comparison. Due to this statistical choice, we could find significant changes in the majority of the investigated parameters, with some of them shared by both young AD and old WT mice, yet further experiments are required to support our hypotheses and possibly establish which of these changes is the most informative of anticipated aging. Furthermore, using typical AD biomarkers (Aβ accumulation, reactive astrocytes, and dystrophic neurons), we did not find close similarities between young AD and aged WT mice. We cannot exclude that in the latter, other changes in astrocytes and microglia as well as subthreshold amount of Aβ or other products of the APP processing may be sufficient to induce the observed network alterations [101,102].

## 5. Conclusions

The aging perspective underlying this study allows us to suggest that specific features of the aging process, in terms of loss of total oscillation power, cortico-hippocampal coupling and functional connectivity in the SO interval, occur in the early stages of AD. Furthermore, hyperexcitability and alterations of the UP-state characterize the AD phenotype from its inception, regardless of Aβ production but aggravated by plaque seeding. We also hypothesize that the aging process itself share striking similarities with the advancement of amyloidosis/gliosis, which may explain convergences in MCI-related network changes. Finally, the fact that middle-aged but not old AD mice show recovery from Low/High power imbalances indicates that more effort should be made to investigate what kind of endogenous mechanisms allow for partially compensating or delaying AD. 

## Figures and Tables

**Figure 1 cells-11-00238-f001:**
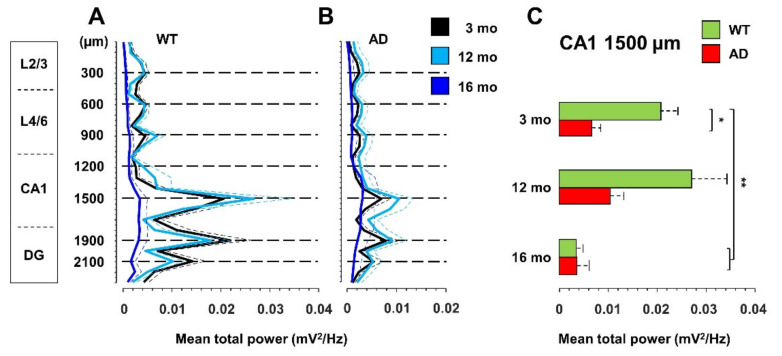
Total power is reduced in young AD and old WT mice. Total power (mean + SEM) profiles of LFP signals, recorded with a linear probe that crosses the PPC and HPF in WT (**A**) and AD (**B**) mice at different ages. The dotted horizontal lines indicate the depths of the seven channels used for all the subsequent analyses. (**C**) Histogram of the total power (mean + SEM) measured at CA1 *sr-lm* (1500 µm) * *p* < 0.05; ** *p* < 0.01, numbers of mice: WT 13, 10, 7 and AD 16, 10, 7 at 3, 12 and 16 months of age, respectively.

**Figure 2 cells-11-00238-f002:**
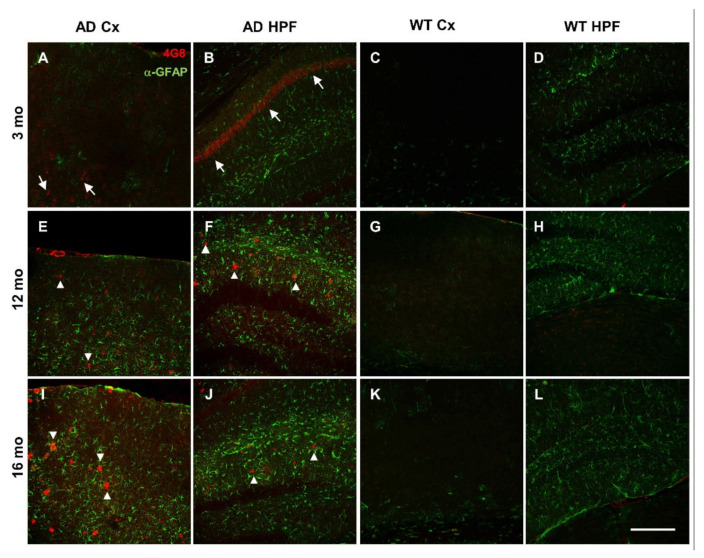
Plaque deposition and astrogliosis increase in AD mice with age. (**A**–**L**) Representative confocal images of immunostaining for APP/Aβ with 4G8 (red) and for astrogliosis with anti-GFAP (green) antibodies, of cortical (Cx) and hippocampal (HPF) regions from coronal slices of 3- (**A**–**D**), 12- (**E**–**H**), and 16- (**I**–**L**) month-old AD and WT mice (20x, scale bar, 200 µm). Arrows indicate 4G8 intraneuronal staining; arrowheads indicate 4G8 staining of Aβ plaques. Confocal images of the entire slices are shown in Appendix A with quantification in Appendix A.

**Figure 3 cells-11-00238-f003:**
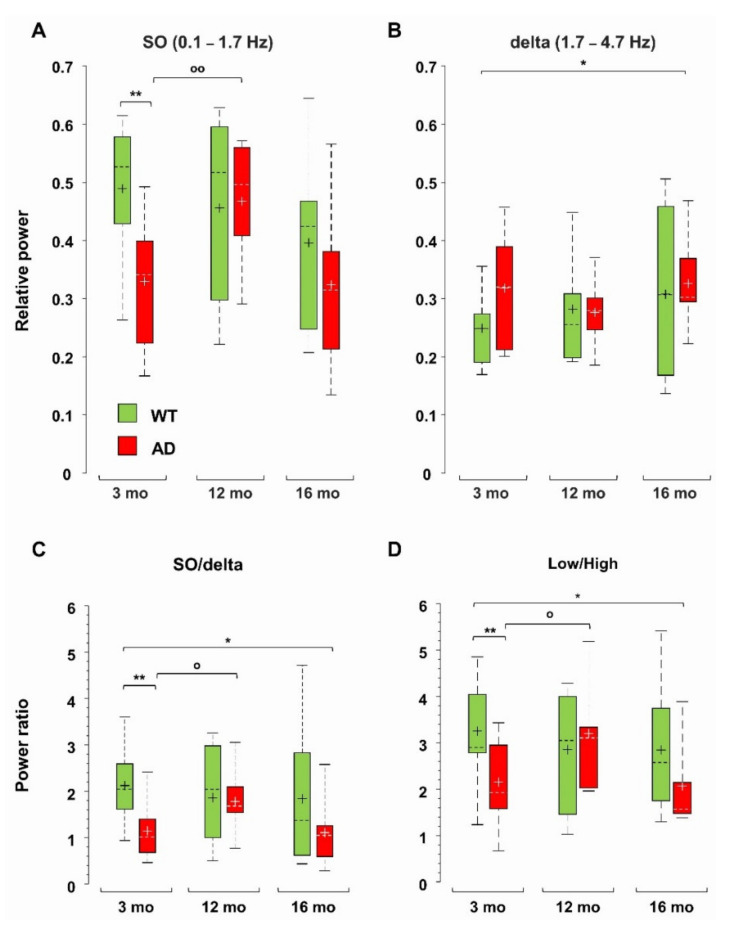
Loss of hippocampal SO relative power and power imbalances characterize young AD mice. Boxplots of the relative power of SO (0.1–1.7 Hz) (**A**) and delta (1.7–4.7 Hz) waves (**B**) in CA1 (*sr-lm*, 1500 µm) of WT and AD mice. At each frequency band, the relative power is the percentage of the total power at the specific depth. Boxplots of the power ratios between SO and delta waves (**C**), and between Low (0.1–4.7 Hz) and High (4.7–190 Hz) frequency bands (**D**), * *p* < 0.05; ** *p* < 0.01). In 12-month-old AD mice, there is a significant increase in SO (**A**) and SO/delta ratio (**C**) versus 3-month-old AD mice, ^o^ *p* < 0.05; ^oo^ *p* < 0.01.

**Figure 4 cells-11-00238-f004:**
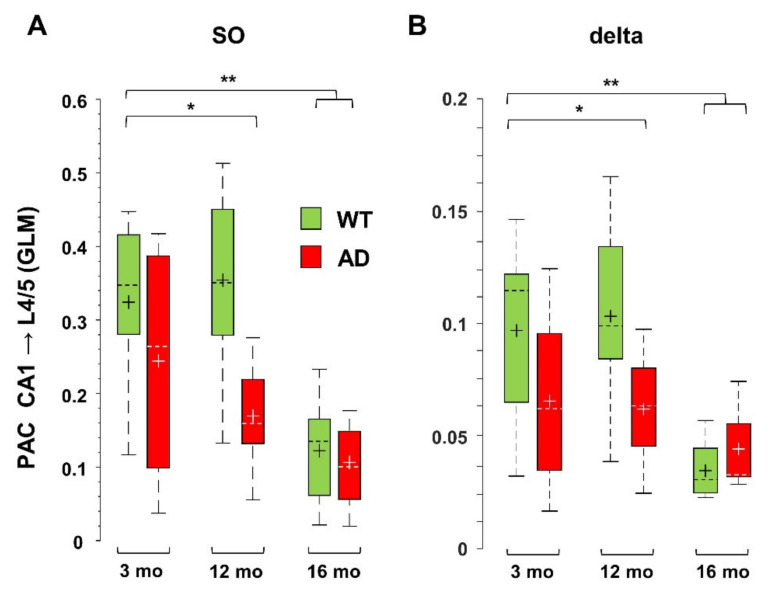
PAC based on SO and delta waves is reduced in middle-aged and old AD as well as in old WT mice. PAC, i.e., the coupling between amplitude of low frequencies and phase of higher frequencies, is measured by the GLM index, as described in Appendix A. Significant reduction was found for PAC occurring between SO (**A**) or delta (**B**) bands in CA1 (*sr-lm*, 1500 µm) and FG in L4/5 (600 µm) of the PPC, * *p* < 0.05; ** *p* < 0.01.

**Figure 5 cells-11-00238-f005:**
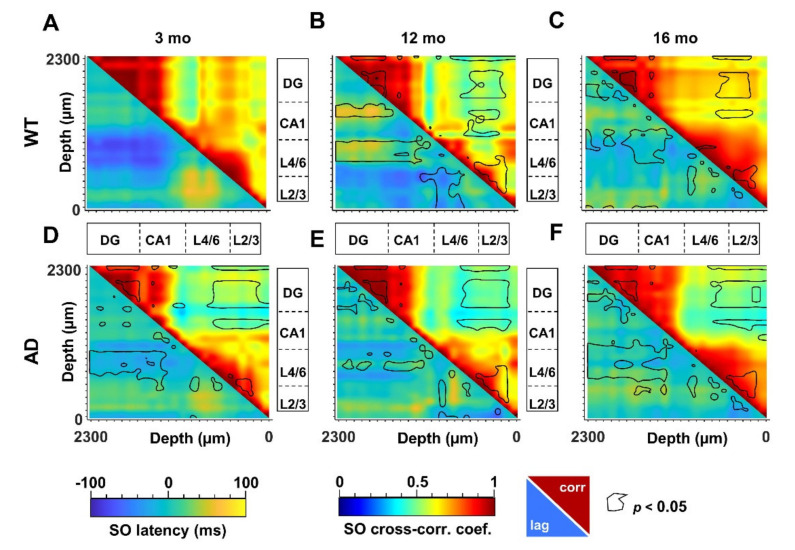
Altered SO cortico-hippocampal connectivity in young AD and middle-aged WT mice. SO connectivity was measured in terms of maximal cross-correlation coefficients and latencies of the instantaneous SO amplitude, as described in Appendix A. Matrices of cross-correlation coefficients (**upper–right**) and latencies (**lower–left**) for 3-, 12- and 16-month-old WT (**A**–**C**) and AD (**D**–**F**) mice were obtained by comparing each recording channel with all the other channels. The matrices report the areas with significant changes with respect to 3-month-old WT mice (*p* < 0.05, black line).

**Figure 6 cells-11-00238-f006:**
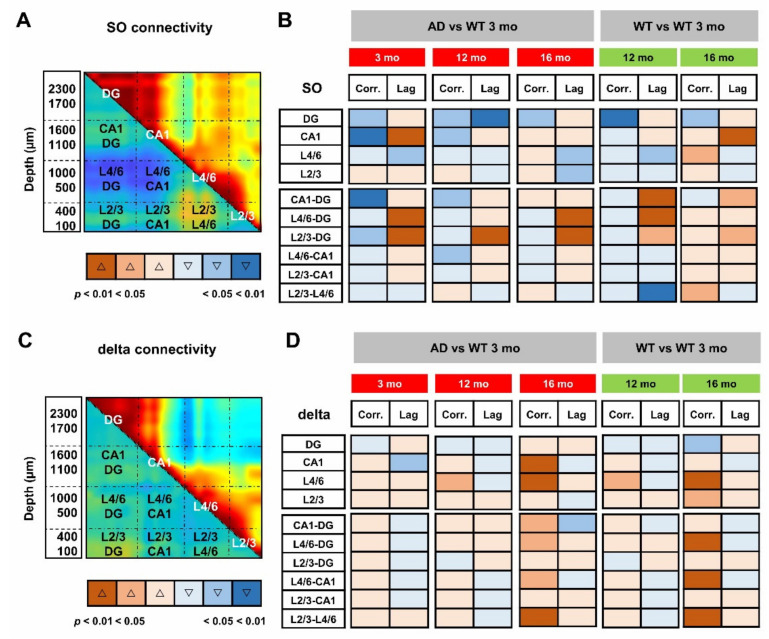
Loss of SO but not delta connectivity anticipates aging in young AD mice. For quantitative analyses of regional changes, maximal cross-correlation coefficients and latencies of each mouse were averaged within (intraregional) and between (cross-regional) regions, according to the scheme overlaid to the matrices of SO (**A**) and delta (**C**) connectivity in 3-month-old WT mice. (**B**,**D**) Synoptic views of the regional changes occurring in maximal cross-correlations and latencies at different ages in AD and WT mice. Warm and cold colors indicate increase and decrease, respectively. The color intensity reflects the level of statistical significance. Changes in regional cross-correlation coefficients and latencies are shown as boxplots in Appendix A.

**Figure 7 cells-11-00238-f007:**
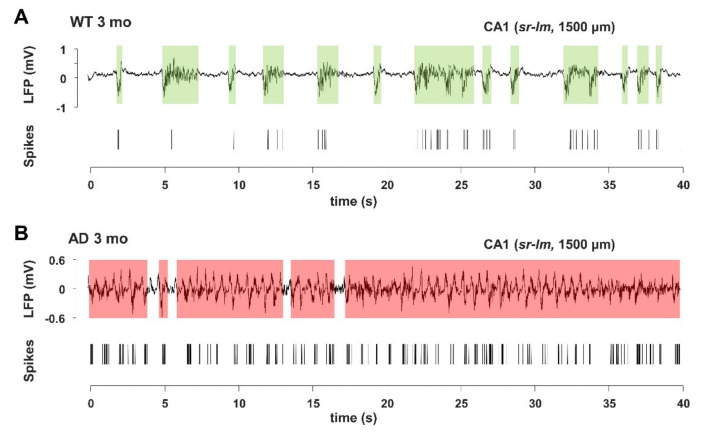
Young AD mice show increase in UP-state duration and spike activity. (**A**,**B**) Representative LFP (upper panels) and spike (lower panels) traces, also indicating the UP-state durations (shaded areas), measured in CA1 (*sr-lm*, 1500 µm) of a 3-month-old WT (**A**) or AD (**B**) mouse, following the method shown in Appendix A.

**Figure 8 cells-11-00238-f008:**
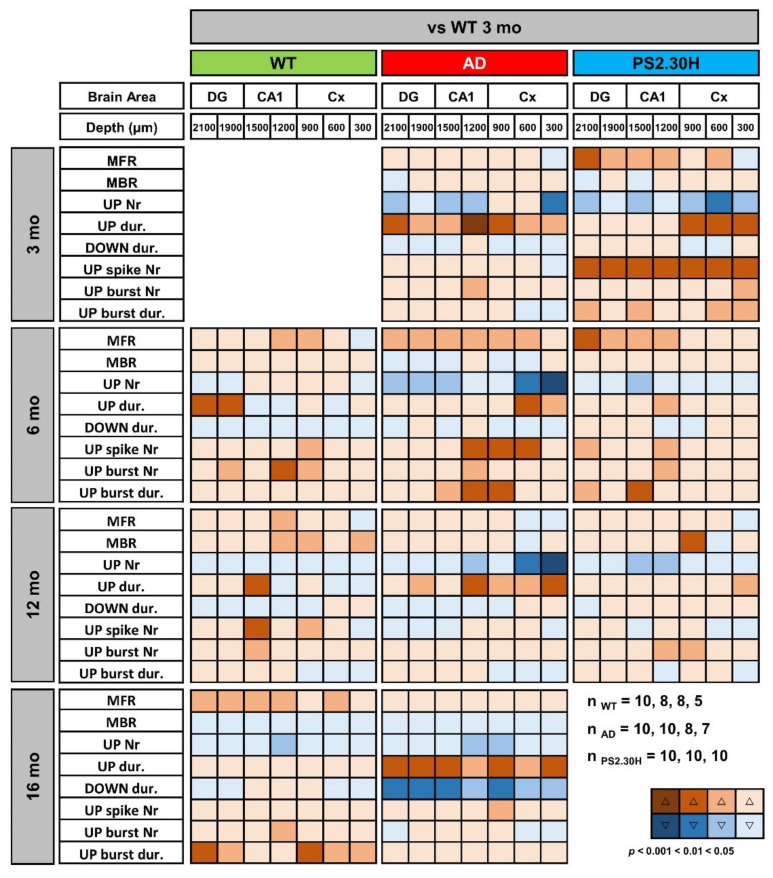
Alterations in UP-DOWN states and spiking activity are shared by mice expressing the mutant PS2. Synoptic views of cortical and hippocampal changes in UP-DOWN states and spiking activity of 3-, 6-, 12- and 16-month-old WT and AD mice, versus 3-month-old WT mice. The same parameters are presented also for the single transgenic PS2.30H mouse line, expressing only the mutant PS2-N141I. Warm and cold colors indicate increase and decrease, respectively. The color intensity reflects the level of statistical significance. With respect to previous analyses, few mice were discarded because of spike artifacts in the selected band region (0.3–3 kHz); *n* indicates the number of mice used for each genotype at 3, 6, 12 and 16 months of age; no mice were available for the16-month-old PS2.30H cohort. MFR, Mean Firing Rate; MBR, Mean Bursting Rate.

**Figure 9 cells-11-00238-f009:**
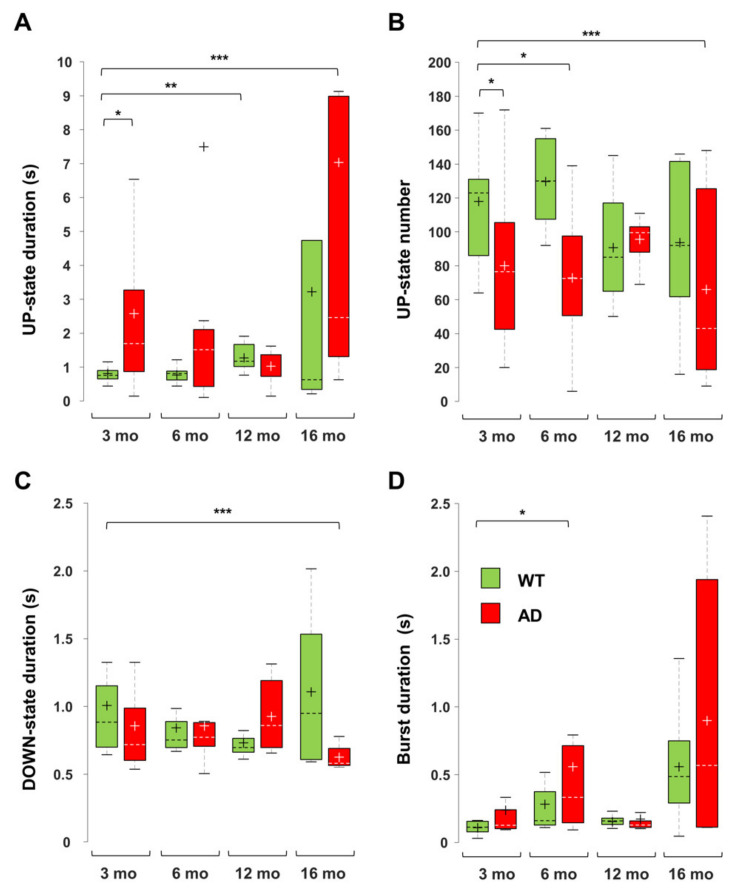
Changes in UP-DOWN states and burst duration in AD and WT mice across ages. Boxplots of UP-state duration (**A**), UP-state number (**B**), DOWN-state duration (**C**) and burst duration (**D**), measured in CA1 *sr-lm* (1500 µm) of AD and WT mice at different ages (* *p* < 0.05; ** *p* < 0.01; *** *p* < 0.001). The numbers of mice per genotype and age cohort are those indicated in Figure 8.

## Data Availability

All data are available upon request.

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
