# Peer review of "Accelerated Aging Characterizes the Early Stage of Alzheimer’s Disease"

_cells, 2022, doi:10.3390/cells11020238_

Round 1
Reviewer 1 Report
Accelerated aging characterizes the early stage of Alzheimer’s Disease is an interesting study.
Authors must make amendments to the statistical data analysis.
1) Data were expressed as mean ± SEM - Be specific about which data.
2) Significance was evaluated by the non-parametric Kruskal-Wallis test for all age cohorts versus 3-month-old WT mice. Where the Kruskal-Wallis test resulted in the existence of a pair of different populations, differences between means were tested with the Mann-Whitney Rank Sum test - This not the correct way of writing this down (text in red).
3) Boxplots show median, mean, the 25th and 75th percentiles of the distributions; whiskers indicate the upper and lower extremes - Does a box plot represent means?
4) Unless otherwise specified, all the reported analyses were based on the following numbers of mice: 3, 10, 7 for WT, and 16, 10, 7 for AD at 3, 12 and 16 months of age, respectively within each group - It should be 13 instead of 3.
5) P values - The way P values have been indicated is not correct. Exact P values should be written, keeping the significance at 0.05. This should be amended throughout the manuscript.
Author Response
Reviewer 1:
Authors must make amendments to the statistical data analysis.
Reply: Yes, we have done it, please see below.
R1.1 Data were expressed as mean ± SEM - Be specific about which data.
Reply: All data in the text are expressed as mean ± SEM. We have now clarified this point in the chapter: Data presentation and statistical analysis.
R1.2 Significance was evaluated by the non-parametric Kruskal-Wallis test for all age cohorts versus 3-month-old WT mice. Where the Kruskal-Wallis test resulted in the existence of a pair of different populations, differences between means were tested with the Mann-Whitney Rank Sum test - This not the correct way of writing this down
Reply: We have now changed the text to better explain the statistical approach. In particular, the intra-age statistical analysis, which consisted in comparing the parameters investigated between all the age cohorts (i.e. 3, 6, 12 and 16 months), was performed using the non-parametric Kruskal-Wallis test. The intra-age pairwise comparison, on the other hand, aimed to compare the parameters investigated between the control group (i.e., 3-month-old WT) and each group of AD mice investigated separately (i.e., B6.152H or PS2.30H), at each age. In this case, the non-parametric Wilcoxon rank sum test was performed. The text has been changed accordingly to lines 113-117.
R1.3 Boxplots show median, mean, the 25th and 75th percentiles of the distributions; whiskers indicate the upper and lower extremes - Does a box plot represent means?
Reply: Where possible, we have adopted boxplots to show the main statistical parameters that characterize the distributions investigated, practically in most of the figures. In the revised version of Data Presentation and Statistical Analysis we better explain the boxplots and their symbols. See lines 111-113
R1.4 Unless otherwise specified, all the reported analyses were based on the following numbers of mice: 3, 10, 7 for WT, and 16, 10, 7 for AD at 3, 12 and 16 months of age, respectively within each group - It should be 13 instead of 3.
Reply: Yes, it was a mistake. See the corrected text at line 120.
R1.5 P values - The way P values have been indicated is not correct. Exact P values should be written, keeping the significance at 0.05. This should be amended throughout the manuscript.
Reply: There are two ways of indicating significance: p <... and p = ..., and both are equally found in published articles. We chose the first mode because it increases the readability of the manuscript; in the revised version we opted for the second, as suggested by the Reviewer, because it makes the data presentation more rigorous.
Reviewer 2 Report
Introduction: “senile dementia” is an unspecific and old-fashioned term from a clinical point of view. Please correct or revise this concept in the intro according to the new nomenclature (normal aging vs mild and major neurocognitive disorder; MCI - Dementia).
Regarding the sentence related to reference 30, please include original research about CFC analysis in AD. In addition, I suggest to introduce the concept of PAC as a biomarker of early AD or in MCI in that section.
Methods: it was used female transgenic mice (PS2APP) as a model of AD. Considering the effects of hormones in neural function during the electrophysiological recordings, how do you control the estrus cycle in those animals?.
Urethane affects oscillatory activity in the hippocampus (i.e. theta band). This kind of anesthesia affected your results?.
Results:
In figure 2. Please include a quantitative method of digital image analysis in order to compare differences in the immunohistochemistry according to the level of fluorescence. Include statistics and representative graphic bars. This is very important to support the subsequent associations with the electrophysiological recordings (i.e. amyloidosis-gliosis and SO connectivity).
How was determined this range of 4.7-190 Hz as high-frequency bands?. Some authors consider “high-frequency oscillations” above the gamma band (up to 30 Hz). PMID: 22449727
Regarding the following sentence: “In young and old (but not middle-aged) AD mice, there is a significant increase in the relative power of the theta band but no significant change is found in WT mice across ages” (supplem Figure 3A). In the supplementary Figure 3A, it is not clear to me that increase in the theta band at 16 months in AD animals in comparison with the same AD animals at 3 or 12 months. This apparent increase is only evident in comparison with 3 months WT mice. Please clarify in the main text.
Using the same kind of analysis from figure 6, it is possible to include some direct comparisons between AD 12 mo Vs WT 12 mo, and AD 16 mo Vs WT 16 groups?. It would be included as supplementary material.
In figure 8B and 8D, what explains this higher variability in the UP-state and burst duration in the AD 16 mo group?. Some animals have epileptiform activity and increase in excitability but there are many other that doesn’t have it?. Do those outliers animals have any particularity in the immunohistochemistry?.
Do you consider any analysis of high-frequency ripple oscillations (around 150-200 Hz)? Ripples and HFO have been associated with epileptiform activity and recently with cognitive function.
Discussion:
Include more references about PAC analysis in AD models (or models of acute amyloidosis) in order to compare with your results: PMID: 23773058; PMID: 33381164; PMID: 25999827.
You consider that low/high power imbalance is a biomarker of early AD. There is any clinical study in patients in a prodromal stage of AD (MCI, APOE epsilon 4 carriers) or in early AD, that have a similar imbalance you found in animals? (i.e. imbalance in theta/delta/gamma bands by EEG).
In addition to the inflammatory changes and calcium imbalance, there are many other mechanisms that explain the differences between normal aging Vs AD? (i.e. autophagy, metabolic and oxidative stress, mitochondrial dysfunction, differences in neurotrophic factor production- BDNF levels, synaptic and neurotransmitter dysfunction).
Reference 80, is about GABA B receptors in Down syndrome. GABA B- GIRK channels response is also affected in models of AD.
Regarding “the early Low/High power imbalance is recovered at 12 months of age”. Could you include any hypothesis about the specific compensatory mechanism?
Author Response
Reviewer 2
R2.1 Introduction: “senile dementia” is an unspecific and old-fashioned term from a clinical point of view. Please correct or revise this concept in the intro according to the new nomenclature (normal aging vs mild and major neurocognitive disorder; MCI - Dementia).
Reply: The Reviewer is right. In the revised version we have removed this definition and simply used the term AD, see line 35.
R2.2 Regarding the sentence related to reference 30, please include original research about CFC analysis in AD. In addition, I suggest to introduce the concept of PAC as a biomarker of early AD or in MCI in that section.
Reply: In the original version the reference 30 was simply included to introduce CFC. In the next sentence, we mistakenly omitted studies on AD models, citing only references from human studies. As suggested by the Reviewer, we have now changed the sentence and introduced references on AD mouse models: PMID: 33381164; PMID: 25999827. The PMID reference: 23773058 was already present (old ref. 23 now ref. 24); see lines 64-65. We have also included a new reference regarding the relevance of PAC in supporting the formation of new memories in the human hippocampus (Lega et al. 2016) as well as other references on this issue (see new refs 32-37), as requested by the Reviewer 3.
References:
Already present:
Goutagny et al. 2013. Alterations in hippocampal network oscillations and theta–gamma coupling arise before Ab overproduction in a mouse model of Alzheimer’s disease PMID: 23773058
New:
Kalweit et al 2015. Acute intracerebral treatment with amyloid-beta (1–42) alters the profile
of neuronal oscillations that accompany LTP induction and results in impaired LTP in freely
behaving rats doi: 10.3389/fnbeh.2015.00103
Gauthier-Umana et al. 2020. Acute Effects of Two Different Species of Amyloid-β on Oscillatory Activity and Synaptic Plasticity in the Commissural CA3-CA1 Circuit of the Hippocampus https://doi.org/10.1155/2020/8869526
Lega, B.; Burke, J.; Jacobs, J.; Kahana, M.J. 2016. Slow-Theta-to-Gamma Phase-Amplitude Coupling in Human Hippocampus Supports the Formation of New Episodic Memories. Cereb. Cortex 2016, 26, 268–278. doi: 10.1093/cercor/bhu232
R2.3 Methods: it was used female transgenic mice (PS2APP) as a model of AD. Considering the effects of hormones in neural function during the electrophysiological recordings, how do you control the estrus cycle in those animals?
Reply: We have not checked the estrus cycle and we agree that it could be a factor of variability, at least for some parameters, not yet defined. However, during the research period, several mice were studied in multiple sessions and for each genotype/age group, mice from 3 to 6 different litters were obtained, possibly preventing bias by randomizing the estrus cycle. Furthermore, in Gurevicius et al. 2013, no correlation was found between estrus cycle and EEG/LFP recordings in different brain regions of AD mice. We have now introduced ref. 47.
Gurevicius, K., Lipponen, A., Tanila, H., 2013. Increased cortical and thalamic excitability
in freely moving APPswe/PS1dE9 mice modeling epileptic activity associated with Alzheimer’s disease. Cereb. Cortex 23, 1148e1158. doi: 10.1093/cercor/bhs105.
R2.4 Urethane affects oscillatory activity in the hippocampus (i.e. theta band). This kind of anesthesia affected your results?
Reply: This is certainly a critical aspect. On the one hand, anesthesia has some advantages, avoiding the more common warning of awake animals, on the other hand it introduces alterations in the brain state that cannot be easily controlled. Urethane anesthesia was chosen because it induces a type of anesthesia reminiscent of sleep, where slow oscillations dominate; moreover, it is one of the most used anesthetics among in vivo studies on plasticity, in particular those with AD mouse models (Lacefield et al., 2012, Marchetti and Marie, 2011, Scott et al., 2012). The fact that we found increased theta relative power in young AD mice, as found in EEG studies (ref. 5) of “Decliner” to AD, in the absence of anesthesia, likely indicates that this band is not primarily affected by this anesthetic. Though experiments with awake animals are preferable, experiments under anesthesia, similar to other ex vivo procedures, such as isolated hippocampi or brain slices, can provide reliable data that could address new problems and help plan experiments in awake or asleep subjects.
In addition to urethane, we have introduced anesthesia with ketamine/xylazine, a procedure that allows to maintain a very stable and prolonged level of anesthesia, which also reduces the death of mice from 50% to 10%. As originally described in Materials and Methods (now in Supp. Materials in the revised version), a preprocessing step was introduced to ensure that all animals were stable during the acquisition procedure. This unbiased procedure identifies the stable heart rate and respiratory windows in which analyses were performed.
Results:
R2.5 In figure 2. Please include a quantitative method of digital image analysis in order to compare differences in the immunohistochemistry according to the level of fluorescence. Include statistics and representative graphic bars. This is very important to support the subsequent associations with the electrophysiological recordings (i.e. amyloidosis-gliosis and SO connectivity).
Reply: In our previous study (Fontana et al. '17), we found no evidence of amyloidosis in WT and PS2.30H mice up to 12 months of age. In that study we show that, in PS2APP mice, plaques begin to appear at 6 months, consistent with this type of AD model (Richards et al. '03; Ozmen et al '09). We also showed high magnification images of probable iAb (4G8 staining) in 3-month-old AD mice (Fontana et al. '17). In the new manuscript we have included only representative images of triple series of 3-, 12- and 16-month-old WT and AD mice, processed simultaneously. As requested by the Reviewer, the quantification is now provided in Supp. Fig. 2D, E. The results are consistent with previous studies, which show that amyloidosis and gliosis reach a plateau in older AD mice, while amyloidosis is not evident in WT mice up to 16 months of age. From Fig. 2 and Sup. Fig. 1, it appears that intraneuronal 4G8 (red) staining is observable in CA1 and cortex of 3-month-old AD mice and becomes weaker at 12 and 16 months, but with the parallel appearance of large plaques. For the sake of clarity, we have added arrows in Fig. 2 and a new Supp. Fig. 2A-C showing higher magnification images of 4G8 staining, both inside and outside the neurons of AD mice at 3, 12 and 16 months. We also addressed the iAb issue in AD and WT mice by comparing 4G8 and MScA1 staining, with the former also staining APP and CTFs. Both antibodies label the neuronal bodies of 1-month-old AD mice, as already shown in Supp. Fig. 8 (now Supp. Fig. 11). In the novel Supp. Fig. 12, we also show MScA1 staining of neuronal bodies in AD mice of 3, 12 and 16 months, but not clear evidence of staining in WT mice (the latter at increased brightness/contrast). Since MScA1 did not show accumulation of iAb in aging WT mice, we shifted our investigation towards inflammation markers, such as co-localization of dystrophic neurons and activation of microglia as already shown in Supp. Fig. 9 (now Supp. Fig. 13), but without obtaining evidence of this type of reactivity in WT mice up to 16 months, even when the latter are imaged at increased brightness/contrast, as now indicated in the legend of the Figure.
Given that alterations in SO connectivity appear with similar characteristics in 3-month-old AD and 12-month-old WT mice, despite large differences in amyloidosis (the total level of Ab42 is approximately 100 times greater in AD mice, Fontana et al. ' 17), and without significant differences in gliosis (see new Supp. Fig. 2F, G), we concluded that it was not possible to find a clear association with these parameters, at least using these tools (4G8/MsCA1, anti-GFAP, -Iba1, -Lamp1). While we can reasonably rule out the involvement of Ab42 and CTFs as a role in the aging process of 12-month-old WT mice (see also general comments to Reviewer 3, R3.4 and R3.10), we recognize that the use of other markers of microgliosis and/or astrogliosis would allow to deepen this theme, as already stated in the original Discussion, see lines 546-552.
R2.6 How was determined this range of 4.7-190 Hz as high-frequency bands?. Some authors consider “high-frequency oscillations” above the gamma band (up to 30 Hz).
Reply: We agree with the Reviewer that the choice of dividing the frequency bands, to which we refer, into two large macro categories, Low (0.1÷4.7 Hz) and High (4.7÷190 Hz) bands sounds arbitrary. In fact, this happens frequently, as researchers are motivated by a specific interest or by previous analyses. For example, in a study describing LFP recordings in human cortex (PMID: 24921388), frequencies below 5 Hz are classified as a "low" band (as in our study) to be later compared to the mid, high and fast. Under our experimental conditions, due to anesthesia, the Low band represents approximately 70% of the total LFP power recorded at each cortical level. With our focus on this low range, we have arbitrarily decided to rate this range against all other bands, defined simply as the High range. Our study also provides a detailed analysis of the specific bands within them (theta, beta, SG, FG and epsilon) and the power ratio of the Low/High band simply provides additional information, without masking other relevant data.
János A Perge et al 2014 J. Neural Eng. 11 046007
Reliability of directional information in unsorted spikes and local field potentials recorded in human
motor cortex PMID: 24921388
R2.7 Regarding the following sentence: “In young and old (but not middle-aged) AD mice, there is a significant increase in the relative power of the theta band but no significant change is found in WT mice across ages” (supplem Figure 3A). In the supplementary Figure 3A, it is not clear to me that increase in the theta band at 16 months in AD animals in comparison with the same AD animals at 3 or 12 months. This apparent increase is only evident in comparison with 3 months WT mice. Please clarify in the main text.
Reply: As mentioned in the Material and Methods (line 116) and in the Discussion (line 427), we focused only on the significant differences compared to the 3-month-old WT mice, which represent the healthy-young condition. Other types of confrontation have been left aside, either by small size or by irrelevance. Specifically, in the Supp. Fig. 3A (now Supp. Fig. 4A), the only significant difference is between the relative theta power of the 3- and 16-month-old AD mice compared to the 3-month-old WT mice, no other significance was found with the other groups.
R2.8 Using the same kind of analysis from figure 6, it is possible to include some direct comparisons between AD 12 mo Vs WT 12 mo, and AD 16 mo Vs WT 16 groups?. It would be included as supplementary material.
Reply: These analyses were also carried out. They have been omitted, due to the large number of figures in this manuscript. These graphs are now shown in the new Supp. Fig 9. At a first inspection, the comparisons by age group show no differences or even changes of opposite sign, compared to the changes observed in the young AD mice. Furthermore, due to the parallel decline of WT mice with aging, the AD changes are partially masked. Comments on these results are now present in the results session at lines 311-316.
R2.9 In figure 8B and 8D, what explains this higher variability in the UP-state and burst duration in the AD 16 mo group? Some animals have epileptiform activity and increase in excitability but there are many other that doesn’t have it?. Do those outliers animals have any particularity in the immunohistochemistry?
Reply: In boxplots, the size of the box gives an idea of the distribution of the data, that is, the data that falls between the first and third quartiles (in these figures there are no outliers). The choice of presenting the data as a boxplot allows us to visualize the variability of the mice with respect to the parameter under examination. For example, in Fig. 4A young AD mice have the greatest variability, suggesting perhaps that, for SO PAC, the conversion between "healthy" and "unhealthy" subjects occurs as early as 3 months for some of them; due to this stratification, significance is not reached at 3 months but only occurs at (6), 12 and 16 months.
As for Fig. 8 in detail, both WT and AD mice show greater dispersion in old age, for all parameters except the duration of the DOWN-state (Fig. 8C). This is consistent with an equally greater dispersion of data for the relative power of the epsilon band, the highest frequency range considered here (new Supp. Fig. 4D). Overall, these parameters are related to hyperexcitability and convulsive-like events. Indeed, in humans, carriers of FAD-related PS2 mutations have a higher incidence of seizures (Jayadev '10, ref. 71), as already reported (see lines 391-393), and PS2.30H mice occasionally exhibit seizures. Even in older WT mice, there is a wide distribution of these parameters, again suggesting a possible stratification towards a “hyperexcitable” AD-like phenotype. However, the fact that old AD, but not WT mice show a significant reduction in the duration of the DOWN-state (Fig. 8C), in the absence of data dispersion, clearly indicates that the AD phenotype is much stronger and dispersion is not due to inaccuracy of the data but to different distributions.
Unfortunately, also because of the many lockdown periods, IHC data came from a set of mice different from that used for electrophysiological recordings.
R2.10 Do you consider any analysis of high-frequency ripple oscillations (around 150-200 Hz)? Ripples and HFO have been associated with epileptiform activity and recently with cognitive function.
Reply: In our study, we analysed the epsilon band (90÷190 Hz), which includes ripple oscillations; we believe that narrowing the frequency band between 150÷190 Hz will not significantly change the results. In fact, we also performed PAC analysis of SO (and delta waves) with the epsilon band, both to and from the PPC and HPF; the results showed a similar loss of SO-PAC, even starting at 6 months of age. We have now included this data in Supp. Fig. 5 as they reinforce the concept that SO-dependent PAC, especially that with FG, is a more sensitive biomarker than that based on delta waves. See also lines 234-236.
Discussion:
R2.11 Include more references about PAC analysis in AD models (or models of acute amyloidosis) in order to compare with your results: PMID: 23773058; PMID: 33381164; PMID: 25999827.
Reply: We thank the Reviewer for the suggestions. As mentioned in R2.2 we changed the text and inserted the new references.
R2.12 You consider that low/high power imbalance is a biomarker of early AD. There is any clinical study in patients in a prodromal stage of AD (MCI, APOE epsilon 4 carriers) or in early AD, that have a similar imbalance you found in animals? (i.e. imbalance in theta/delta/gamma bands by EEG).
Reply: We agree with the Reviewer that this issue is of the utmost interest. Indeed, we had already included a comment in this direction (see Discussion lines 563-566 and ref. 5) “Concerning the power imbalances, it is worth noting that, in humans, an increase in theta relative power was detected by the quantitative electroencephalogram (QEEG) in normal elderly subjects, subsequently identified as “Decliners” to dementia with respect to “Non-Decliners”. The increase in relative theta power (3.5-7.5 Hz) is consistent with our data, yet no changes were reported in the lowest frequency range considered in that study (1.5-3.5 Hz). It is interesting that also in mice under anesthesia we could find a significant increase in relative theta power at the early stage of disease. Indeed, increased theta power is now considered an early biomarker of AD (Museaus et al. ‘18 DOI: 10.3233/JAD-180300; Smailovic et al. ’18, PMID: 29245058).
As far other studies in humans, by using EEG recordings in sleeping subjects, it was demonstrated that the loss of slow wave activity (SWA, <1 Hz) but not delta waves (1-4 Hz) predicts levels of Aβ aggregation as measured by PET (Mander et al., 2015, doi:10.1038/nn.4035), see also refs. 51 & 52 in our work. We have now further stressed this issue, in the Discussion referring to these studies, see lines 571-573.
R2.13 In addition to the inflammatory changes and calcium imbalance, there are many other mechanisms that explain the differences between normal aging Vs AD? (i.e. autophagy, metabolic and oxidative stress, mitochondrial dysfunction, differences in neurotrophic factor production- BDNF levels, synaptic and neurotransmitter dysfunction).
Reply: We agree with the Reviewer's comment, we forgot to mention autophagy, as well as differences in neurotransmitters and synaptic dysfunction. In fact, we have partially mentioned some of the possible mechanisms, also considering the fact that a description of autophagic, metabolic, energetic, and redox dysfunctions, linked to the expression of PS2 in AD and aging, is already present in our works (old refs. 67, 76, 77 now refs. 82,90,91). We have introduced new refs 70, 81 (see also comments at line 538).
R2.14 Reference 80, is about GABA B receptors in Down syndrome. GABA B- GIRK channels response is also affected in models of AD.
Reply: We believe that there was a misunderstanding: reference 80 deals with Down-state transition in WT mice not in Down syndrome (Hay, Y.A.; Deperrois, N.; Fuchsberger, T.; Quarrell, T.M.; Koerling, A.-L.; Paulsen, O. Thalamus mediates neocortical Down state transition via GABAB-receptor-targeting interneurons. Neuron 2021, 109, 2682-2690.e5, doi:10.1016/j.neuron.2021.06.030.).
The reason to include that reference was based on the consideration that, at variance with isolated organs and slices, the search for the mechanisms responsible of UP/DOWN-state alterations in vivo requires to consider also the connections with other brain regions as the possible cause.
Concerning GABA B-GIRK channels we have now included a new sentence in the Discussion see lines 535-540 and ref. 97.
R2.15 Regarding “the early Low/High power imbalance is recovered at 12 months of age”. Could you include any hypothesis about the specific compensatory mechanism?
Reply: At this stage, we have no clear indication from our data of a possible rescue mechanism for SO, but we simply noted that, at 12 months, some alterations are less marked than at 3 and 16 months. However, it is worth noting that the B6.152H mouse model was recently used in a longitudinal study, in which the same mice were analyzed by micro-PET for amyloidosis and gliosis at different ages (8 to 13 months) (Focke et al. '19, old ref. 61, now 74); behavioral, immunohistochemical and biochemical analyses completed the study. Note that the best cognitive performance was directly associated with the level of microglia activation 5 months earlier (i.e. 8 months). These data suggest that microglia activation has a protective role on later cognitive performance if it occurs early in the disease, possibly acting as a buffer. This aspect of the rescue mechanism has not been discussed as it is still too preliminary for our study.
Reviewer 3 Report
General comment:
The authors approach a relevant question for the field, how aging potentiates AD development and which biomarkers can help predict the conversion from normal to MCI and from MCI to AD. However, the experimental research approach used is based on rare familial experimental models that mechanistically are caused but mutations and relatively unlinked from aging mechanisms precluding direct conclusions. The analysis of the aging cognitive decline is wt mice is the most exciting part. However, there are some overinterpretations in the independence of amyloid production, which are biased and not supported by previous data nor by the qualitative analysis of the poorly presented brain immunostainings.
Specific comments:
Abstract.
The abstract is poorly structured, and the underlying message is unclear regarding the results and conclusions.
For example:
Problem: “…, and most ignore how the aging process affects control mice. “ Solution: “In this work, we addressed this problem by studying the spontaneous brain electrical activity of anesthetized PS2APP (AD) mice.” How does studying PS2APP mice answer the problem of how the aging process affects control mice? The abstracts needs rewriting so that the aims match the results and the approach.
Introduction
Not being knowledgeable in theta-gamma cross-frequency coupling (CFC), I was surprised by such a bold affirmation that it is the phenomenon at the basis of memory coding. After searching for supporting data, I found that there may be confounding studies (https://elifesciences.org/articles/44287). Moreover, it would be essential to explain the CFC mechanism and its basis for memory.
Results:
The result presented in figure 1 that spontaneous brain electrical activity is not different between the AD and the aged 16m old mice is very surprising. What could be the explanation?
In figure 2, the authors analyze the pathological changes with aging and show a representative merged image of amyloid plaques and intraneuronal Abeta. Unfortunately, the merged image does not allow the evaluation of intraneuronal Abeta changes. The intraneuronal Abeta image should be shown separately. Moreover, the Abeta signal should be shown in white or green because the red precludes a comparable visualization of the signal. Notably, Abeta levels in the AD mouse brain are incomparably higher than in control mice, impeding the analysis of the Abeta levels in the control mice brain with age. To observe intraneuronal Abeta in control mice, the brightness and contrast of the Abeta signal need to be adjusted and presented as an additional separate panel, even if in supplementary data. Finally, the lack of quantitative analysis of Abeta or GFAP IHC undermines the significance of these observations.
This reviewer's lack of expertise in oscillations and connectivity changes does not allow for revising the data presented in the other figures.
On page 12, the authors mention being interested in determining causality between PS2, abeta or both. However, it is unclear how analyzing mice overexpressing a mutant PS2 enables the conclusion that the effects observed are related to the function of PS2 alone, since the FADN141I mutation in PS2 is the familial mutation with the most impact on intraneuronal Abeta 42 production ((Ab42/40 >50-fold) even without affecting extracellular Abeta (Sannerud, 2016 Cell). Thus the results need to be interpreted accordingly.
The sentence: “a primary role played by mutant PS2, in the absence of Aβ accumulation and gliosis “ is referring to amyloid plaques and disregards intraneuronal Abeta. The authors are extrapolating their conclusions without providing data and without literature support. This needs to be addressed experimentally.
On page 13, the authors write, “In adult WT mice, we occasionally found iAβ, as defined by 4G8 reactivity (Figs. 2G, H; Supp. Figs. 1D and 9E-L), yet it was not reproduced consistently with MsCA1, likely because 4G8 also detects APP/βCTFs, thus reflecting the large variability in the level of APP expression in mice.“ In the literature, the majority of studies found no differences in APP expression with normal aging. The lack of changes in APP support that the changes in APP processing and not in its expression occur with aging. This alternative explanation needs to be considered by the authors.
Moreover MsCA1 lack of consistency may be due to being an antibody developed against human Abeta and may not recognize mouse Abeta. Could the authors discard this possibility? Have the authors confirmed the absence of signal of MSCA1 in Brains of APP knockout mice? Have the authors used 12F4 that recognizes Abeta 42 in wild-type mice and not in APP knockout mice?
The authors should do the controls mentioned above. Otherwise, conclusions that the connectivity differences found are independent of increased Abeta production and intraneuronal accumulation with aging, please see a recent publication on this issue (Burrinha JCS 2021) cannot be made. Conclusions The conclusion “hyperexcitability and alterations of the UP-state characterize the AD phenotype from its inception, regardless of Aβ production….” is not supported by the data presented in the paper.
Author Response
Reviewer 3
General comment:
The authors approach a relevant question for the field, how aging potentiates AD development and which biomarkers can help predict the conversion from normal to MCI and from MCI to AD. However, the experimental research approach used is based on rare familial experimental models that mechanistically are caused but mutations and relatively unlinked from aging mechanisms precluding direct conclusions. The analysis of the aging cognitive decline is wt mice is the most exciting part. However, there are some overinterpretations in the independence of amyloid production, which are biased and not supported by previous data nor by the qualitative analysis of the poorly presented brain immunostainings.
Reply: We thank the Reviewer for appreciating the relevance of our approach. However, we partially disagree with the criticisms regarding the choice of experimental model, the overinterpretation of the data and the quality of the immunostaining.
As for the experimental model, there are many AD mouse models, among these those based on the overexpression of FAD mutations are the most used and each of them, if properly studied, allows to identify the characteristics of this multifaceted disease. In particular, for many years, we have focused on the PS2APP model, and on the closely related one, the single transgenic model PS2.30H which has allowed us to shed light on PS2, both the mutated and the endogenous one, due to its primary role on Ca2+ homeostasis at the cytosolic, ER and mitochondrial level, as well as on energy balance, oxidative stress and autophagy (see Pizzo et al. 2020, and references therein). In the last decade, the PS2APP model has been widely used in MRI and micro-PET studies for amyloidosis and microgliosis, associated with behavioral assays (old refs. 61-63 now 74-76).
Regarding overinterpretation and quality of immunostaining, the rationale for our approach was based on the following considerations: 12-month-old WT mice share network similarities with 3-month-old AD mice, however amyloidosis/gliosis analyses, performed by IHC reveal no similarities, except for a trend, although not significant, for an increase in GFAP staining in aged WT mice, now quantified in Supp. Figs. 2F,G. In our previous study, we measured the total (soluble and insoluble) amount of Ab42 and Ab40, upon formic acid extraction, by Wako ELISA assays that recognize both human and mouse epitopes. In WT (and PS2.30H) mice, we did not observed an increase in Ab42, up to 12 months (Fontana et al. '17 Supp. Fig. 6). These results are consistent with the fact that the endogenous mouse APP is much less amyloidogenic and much less abundant than the human APP over-expressed in PS2APP mice. Furthermore, staining with MsCA1, an antibody specific to Ab42 (Iulita et al., 2014; Welikovitch et al., 2018, refs 72, 73) did not reveal iAb in WT mice for up to 16 months, but works well in AD mice (see Supp. Figs. 11,12).
We are sorry, but probably the resolution of the pdf images does not allow you to appreciate the quality of our images. In the revised version we have increased it as much as possible. It should also be noted that anti -AbAPP (4G8), -Iba1, -GFAP and -Lamp1 are the most widely used antibodies in the field of AD. In particular, for plaque detection in IHC, 4G8 is considered superior because it recognizes both human and murine Ab equally well, marking both dense and diffuse plaques (cores and halos). In the revised version we have added arrows to highlight the details in the old figures, and new images to visualize the intraneuronal staining with 4G8 (MsCA1) and better appreciate this type of results.
Specific comments:
Abstract.
R3.1 The abstract is poorly structured, and the underlying message is unclear regarding the results and conclusions.
For example:
Problem: “…, and most ignore how the aging process affects control mice. “ Solution: “In this work, we addressed this problem by studying the spontaneous brain electrical activity of anesthetized PS2APP (AD) mice.” How does studying PS2APP mice answer the problem of how the aging process affects control mice? The abstracts needs rewriting so that the aims match the results and the approach.
Reply: We thank the Reviewer for the suggestion to improve the Abstract. We have now modified it accordingly to highlight the novelty of the approach, which is based on the investigation of both aging and progression of AD. See changes to lines 18-19.
Introduction
R3.2 Not being knowledgeable in theta-gamma cross-frequency coupling (CFC), I was surprised by such a bold affirmation that it is the phenomenon at the basis of memory coding. After searching for supporting data, I found that there may be confounding studies (https://elifesciences.org/articles/44287). Moreover, it would be essential to explain the CFC mechanism and its basis for memory.
Reply: We thank the Reviewer for giving us the opportunity to better introduce this issue. However, we don't think that considering CFC at the basis of memory coding is a bold affirmation. With a quick PubMed search using cross-frequency coupling and memory as keywords, we got numerous references that, even on first inspection, clearly demonstrate the importance of CFC in memory processing. Most of them describe the role of CFC in human cognitive performance, both in terms of predictivity (PMID: 25535839); causal (PMID: 33741402) or support (PMID: 33544408; https://doi.org/10.1093/cercor/bhu232) roles; others also focus on animal models (PMID: 33588406; PMID: 32661022).
Unfortunately, in the original version we put only one reference, more centered on the approach. In the revised version we have modified the sentence and selected some other references from below (in bold), addressing CFC and memory encoding. See lines 219-221and references 23, 24, 33-37, 53, 54.
As for the confounding studies, we are aware of the critical aspects of this approach. Indeed, in the past, our group has thoroughly investigated this problem (Rubega et al 2016), demonstrating that the General Linear Model (GLM) is a better and safer estimator of PAC than the modulation index (MI), being less affected by signal noise and the length of the signal chunk. Therefore, this approach has been routinely employed in our works (Fontana et al. '17, Leparulo et al. '20). This information was already present (see Supp. Materials) and explained in detail in Fontana et al. '17. Curiously, the work cited by the Reviewer is also based on GLM.
PubMed search for cross-frequency-coupling and memory:
- Working Memory and Cross-Frequency Coupling of Neuronal Oscillations.
Abubaker M, et al. Front Psychol. 2021. PMID: 34744934
- Corticothalamic phase synchrony and cross-frequency coupling predict human memory formation.
Sweeney-Reed CM, et al. Elife. 2014. PMID: 25535839
- Frontoparietal Beta Amplitude Modulation and its Interareal Cross-frequency Coupling in Visual Working Memory.
Liang WK, et al. Neuroscience. 2021. PMID: 33588001
- Cross-regional phase amplitude coupling supports the encoding of episodic memories.
Wang DX, Schmitt K, Seger S, Davila CE, Lega BC.Hippocampus. 2021 May;31(5):481-492. doi: 10.1002/hipo.23309. Epub 2021 Feb 5.PMID: 33544408
- Causal role of cross-frequency coupling in distinct components of cognitive control.
Riddle J, McFerren A, Frohlich F.
Prog Neurobiol. 2021 Jul;202:102033. doi: 10.1016/j.pneurobio.2021.102033. Epub 2021 Mar 16.PMID: 33741402
- Assessing the Longitudinal Relationship between Theta-Gamma Coupling and Working Memory Performance in Older Adults.
Brooks H, Mirjalili M, Wang W, Kumar S, Goodman MS, Zomorrodi R, Blumberger DM, Bowie CR, Daskalakis ZJ, Fischer CE, Flint AJ, Herrmann N, Lanctôt KL, Mah L, Mulsant BH, Pollock BG, Voineskos AN, Rajji TK. Cereb Cortex. 2021 Sep 14:bhab295. doi: 10.1093/cercor/bhab295. Epub ahead of print. PMID: 34519333.
- Brain network features based on theta-gamma cross-frequency coupling connections in EEG for emotion recognition.
Wang W Neurosci Lett. 2021 Sep 14;761:136106. doi: 10.1016/j.neulet.2021.136106. Epub 2021 Jul 9. PMID: 34252515.
- Odour Retrieval Processing in Mice: Cholinergic Modulation of Oscillatory Coupling in Olfactory Bulb-Piriform Networks.
Ahnaou A, Chave L, Manyakov NV, Drinkenburg WHIM.Neuropsychobiology. 2021;80(5):374-392. doi: 10.1159/000513511. Epub 2021 Feb 15.PMID: 33588406
- Cross-Frequency Phase-Amplitude Coupling between Hippocampal Theta and Gamma Oscillations during Recall Destabilizes Memory and Renders It Susceptible to Reconsolidation Disruption.
Radiske A, Gonzalez MC, Conde-Ocazionez S, Rossato JI, Köhler CA, Cammarota M.
J Neurosci. 2020 Aug 12;40(33):6398-6408. doi: 10.1523/JNEUROSCI.0259-20.2020. Epub 2020 Jul 13.PMID: 32661022
- Slow-Theta-to-Gamma Phase-Amplitude Coupling in Human Hippocampus Supports the Formation of New Episodic Memories.
Lega, B.; Burke, J.; Jacobs, J.; Kahana, M.J. Cereb. Cortex 2016, 26, 268–278. doi: 10.1093/cercor/bhu232
Results:
R3.3 The result presented in figure 1 that spontaneous brain electrical activity is not different between the AD and the aged 16m old mice is very surprising. What could be the explanation?
Reply: We too were surprised by this finding. Since total power loss was found in young AD mice, at the hippocampus level, and subsequently, also at the cortical level, in both AD and WT mice, we interpreted this as a feature linked to accelerated aging. Consistently, total power loss also characterizes young APPSwe and PS2.30H mice, the closest AD mouse models, but not PSEN2-/- mice (Leparulo et al. ‘20), reinforcing the idea that AD is linked to accelerated aging.
R3.4 In figure 2, the authors analyze the pathological changes with aging and show a representative merged image of amyloid plaques and intraneuronal Abeta. Unfortunately, the merged image does not allow the evaluation of intraneuronal Abeta changes. The intraneuronal Abeta image should be shown separately. Moreover, the Abeta signal should be shown in white or green because the red precludes a comparable visualization of the signal. Notably, Abeta levels in the AD mouse brain are incomparably higher than in control mice, impeding the analysis of the Abeta levels in the control mice brain with age. To observe intraneuronal Abeta in control mice, the brightness and contrast of the Abeta signal need to be adjusted and presented as an additional separate panel, even if in supplementary data. Finally, the lack of quantitative analysis of Abeta or GFAP IHC undermines the significance of these observations.
Reply: The purpose of Fig. 2 was not intended to show the presence of iAb, which was better stained by a more specific antibody such as MsCA1 in Supp. Fig. 8 (now Supp. Fig. 11) and the novel Supp. Fig. 12. Fig. 2 and Supp. Fig. 1 were intended to provide a comparison between AD and WT slices with the same antibodies and the same acquisition procedure. In the new Fig. 2, we have now added arrows to indicate where 4G8 intraneuronal staining (probably iAb) is detectable, as well as arrowheads for plaque staining. In the new Supp. Fig. 2A-C and Supp. Fig. 12, we have added new images of 4G8 staining in AD mice. In Supp. Fig. 12, MsCA1 staining in WT mice is shown at higher contrast than in AD mice, but no clear differences were found in WT mice of all ages and no similarity to young AD mice.
R3.5 This reviewer's lack of expertise in oscillations and connectivity changes does not allow for revising the data presented in the other figures.
Reply: Ok
R3.6 On page 12, the authors mention being interested in determining causality between PS2, abeta or both. However, it is unclear how analyzing mice overexpressing a mutant PS2 enables the conclusion that the effects observed are related to the function of PS2 alone, since the FADN141I mutation in PS2 is the familial mutation with the most impact on intraneuronal Abeta 42 production ((Ab42/40 >50-fold) even without affecting extracellular Abeta (Sannerud, 2016 Cell). Thus the results need to be interpreted accordingly.
Reply: The Reviewer is correct when considering the effect of PS2-N141I in human subjects or human FAD fibroblasts (Sannerud et al. '16). When using single transgenic mouse models expressing a human PS1/2 mutation, it should be borne in mind that mouse APP is far less amyloidogenic than human APP. In particular, up to 12 months of age, the PS2.30H line shows neither total accumulation of Ab42 (Fontana et al. '17) nor plaques (unpublished data and L. Ozmen, personal communication). However, the effect of PS2 mutations on other substrates and cellular pathways could be present in single transgenic mice, giving us the ability to distinguish them from amyloidosis. We have previously shown that the Ca2+ dysregulation found in human cell lines and FAD fibroblasts, expressing mutated PS2, is also present in neurons and astrocytes of PS2.30H and PS2APP mice (Kipanyula et al. '12). Furthermore, structural and functional defects in the ER, mitochondria and phagosomes are also found in human FAD fibroblasts expressing mutated PS2 (Greotti et al. '19; Fedeli et al. '19; Pizzo et al. 2020; now refs 70, 82, 86).
On page 12, lines 378-382, we have now added a sentence to better explain the logic of our approach and clear up this rather confusing problem.
R3.7 The sentence: “a primary role played by mutant PS2, in the absence of Aβ accumulation and gliosis “ is referring to amyloid plaques and disregards intraneuronal Abeta. The authors are extrapolating their conclusions without providing data and without literature support. This needs to be addressed experimentally.
Reply: We haven't ignored iAβ, but we doubt it could play a major role in defective SO connectivity. See above R2.5.
R3.8 On page 13, the authors write, “In adult WT mice, we occasionally found iAβ, as defined by 4G8 reactivity (Figs. 2G, H; Supp. Figs. 1D and 9E-L), yet it was not reproduced consistently with MsCA1, likely because 4G8 also detects APP/βCTFs, thus reflecting the large variability in the level of APP expression in mice.“ In the literature, the majority of studies found no differences in APP expression with normal aging. The lack of changes in APP support that the changes in APP processing and not in its expression occur with aging. This alternative explanation needs to be considered by the authors.
Reply: Indeed APP belongs to the genes that are subject to random monoallelic expression (Gimelbrandt et al. '07 doi: 10.1126 / science.1148766; Gui et al. '17 doi: 10.3389 / fgene.2017.00191, ref. 77), and differences in the expression levels of APP, due to mosaicism and somatic recombination, have been suggested among the possible causes of SAD (Lee et al. '18 https://doi.org/10.1038/s41586-018-0718-6; Constantino et al. 2021 https ://doi.org/10.3390/genes12071071). Random monoallelic expression is also present in mice and is likely greater when the mice are kept outbred rather than inbred, as in our case. Indeed, differences in mAPP expression level are clearly visible in 6-month-old WT, PS2KO and PS2.30H mice (see Supp. Fig. 9 by Fontana et al. '17).
R3.9 Moreover MsCA1 lack of consistency may be due to being an antibody developed against human Abeta and may not recognize mouse Abeta. Could the authors discard this possibility? Have the authors confirmed the absence of signal of MSCA1 in Brains of APP knockout mice? Have the authors used 12F4 that recognizes Abeta 42 in wild-type mice and not in APP knockout mice?
Reply: Although developed against human Ab, the MsCA1 antibody works well in mice (old Supp. Fig. 8 and new Supp. Fig. 12) and rats (refs 72, 73). In addition, intraneuronal aggregates of Ab, stained by 4G8, have been described in C57Bl/6 mice from 15 months to 22, but not 12 months of age (Ahlemeyer et al. 2018 DOI 10.3233 / JAD-170923). We see no reason to use APP knockout mice, as the aim was not to demonstrate the validity of different antibodies or to test for the presence of iAb in WT mice over 16 months of age.
R3.10 The authors should do the controls mentioned above. Otherwise, conclusions that the connectivity differences found are independent of increased Abeta production and intraneuronal accumulation with aging, please see a recent publication on this issue (Burrinha JCS 2021) cannot be made. Conclusions The conclusion “hyperexcitability and alterations of the UP-state characterize the AD phenotype from its inception, regardless of Aβ production….” is not supported by the data presented in the paper.
Reply: We do not exclude that even in our WT mice, in older age, we could find an increased reactivity to MsCA1 (or 12F4) compared to young and middle-aged WT mice. We simply state that the rather similar level of MsCA1reactivity, present in 3- and 12-month-old WT mice, but widely different from that of 3-month-old AD mice, hardly explains the network similarities between the latter and 12-month-old WT mice.
Regarding the work cited by the Reviewer (Burrinha et al. (2021), we believe that a comparison with our results is rather difficult. In fact, the vast majority of the data in that work comes from embryonic neuronal cultures: altered APP endocytosis/processing and accumulation of iAb have been observed in primary neuronal cultures from BALB mice, showing aging effects at 28 DIV, but not 21. The correspondence between aging neurons in vitro and neurons in vivo is difficult to establish. Furthermore, it is known that, even in mouse strains, the genetic background largely influences APP processing and Ab deposition (Lehman et al. '03, DOI: 10.1093/hmg/ddg322).
Burrhina et al. (2021) also show IHC data of C57Bl/6 mice at 18 months, an age we were not interested in. Of note, in that work, 12F4 antibody was used to detect iAb in neurons grown from BALB mice but not in situ from old C57Bl/6 mice. In that study, IHC showed alterations in APP endocytosis at 18 months, using anti-EEA1 and anti-APP antibodies. Perhaps 12F4 was not working in old C57Bl/6 mice or its reactivity was below the threshold. Curiously, the control came from 12-month-old mice.
Burrhina's work focuses primarily on the effect of aging on endocytosis and processing of mAPP by immunofluorescence with state-of-the-art approaches. In that work, Western blots of brain homogenates from 18-month-old C57Bl/6 mice show similar mAPP levels, compared to 6 months, but increased levels of CTFs, notably C83/89, but also C99, only in elderly mice. Unfortunately, the levels of AICD and Ab, the direct products of gamma-secretase, were not shown. Increased levels of CTFs suggest increased alpha and/or beta-secretase activity, but also reduced degradation of CTFs. Indeed, these products could have toxic effects, even at the network level (Hamm et al. '17, Sci. Adv. 2017; 3: e1601068; Xu et al. '14 http://dx.doi.org/10.1016 / j.expneurol.2014.12.008). We have already discussed that CTFs could play a role on the network dysfunctions observed in B6.152H mice from 6 months of age, possibly aggravating the AD phenotype (see Discussion Leparulo et al. ‘20, page 15). We have avoided repeating these considerations in the new discussion.
Regarding the problem of accelerated aging, we believe that the reasoning we applied to iAb42 is also suitable for CTFs: the amount present in 3-month-old AD mice - which overexpress hAPP - is far greater than that found in 12-month-old WT mice.
We believe that the conclusion "hyperexcitability and alterations of the UP-state characterize the AD phenotype since its inception, regardless of Aβ production...". is still sustainable being based on the dramatic effects caused by PS2-N141I in PS2.30H mice, compared to WT mice, at multiple levels: excitability (Fig. 9, this work, Fontana et al. '17 and Leparulo et al. '20), Ca2+ homeostasis, autophagic/metabolic pathways, but in the absence of significant production of Aβ42, as previously documented.
Reviewer 4 Report
This study is on spontaneous intracranial EEG signatures and immunohistochemical presentation of mice at a wide range of ages, and their transgenic counterparts carrying one of the dominantly inherited genes of Alzheimer’s disease (presenilin 2). While I think the experimental design is interesting, there are major issues with the presentation of the data and its interpretation. The comments below are made in a page-by-page manner, with the most important ones indicated by an asterisk (*).
- The title of this work makes a bold statement, which is not sufficiently backed up by the data. Specifically, authors interpret the EEG changes seen as “accelerated aging”. However, the current widely accepted view of aging (see e.g., The Hallmarks of Aging, López-Otin et al, Cell 2013) does not include EEG dynamics. Therefore, while there is some consistency in the interpretation that young AD mice resemble old WT mice in their EEG dynamics, the title should reflect much more precisely what was actually seen. This issue is of particular importance in the present study because the canonically accepted hallmarks of aging are unidirectionally progressive over time, whereas the EEG dynamics reported by the authors are reversed at middle age in AD mice.
Secondly, mice do not have Alzheimer’s disease (AD), and the presenilin 2 mutation is present in a small minority of all human AD cases (not important in the late-onset AD, which represents 95% of all cases, and the least common mutation of the familial early onset AD). Therefore, I do not think it is justified that the title speaks about “early stage of Alzheimer’s disease”. Rather, it should include a phrase such as “PS2APP mouse model of Alzheimer’s disease” or “PS2APP mice” or “transgenic mouse model of Alzheimer’s disease” or equivalent, like the previous papers of the authors utilizing the same model have done.
- Materials and Methods/Animal Preparation and Surgery (lines 112-117): please provide the stereotaxic coordinates of the probe.
- Materials and Methods/Signal Acquisition (lines 146): Supplementary Materials do not have any information about the quantitative evaluation of anesthesia level.
- * Results (sections 3.2, 3.3, 3.4): The authors do not report anywhere the low cutoff frequency of their EEG data acquisition system. Moreover, as the Supplementary Figure 2 shows that the PSD has its apex at 1Hz and decreases by almost two orders of magnitude from 1Hz to 0.1Hz, it is evident that the effective low cutoff frequency of the system is around 1Hz. Therefore, the data on slow oscillations (SO) in the frequency band 0.1-1 Hz are not valid.
- * Results (Figure 1): By the power law of neural oscillations, slow frequency oscillations provide the major contribution to the total power over all frequencies. Because of SO recorded in this study seem to be not valid, also the total power including these frequencies should not be used. Also, because of the disproportionate contribution of slow frequencies, the plots in Figure 1 should rather be shown for each frequency band separately.
- * Results (Figure 1): The plot for WT mice at 16 months of age seems as if these mice have almost lost all of their EEG activity across the depth probed. Again, due to power law, this is likely due to a remarkable shift in slow oscillation dynamics. Please show the raw EEG traces at least from CA1 (1500um) and DG (2100um) depths so that the readers can see how does this remarkable difference between 12 and 16 months look in the EEG. The band-specific representation requested in comment 5 above would also make it visible whether the change seen is across all modes of neural activity, or whether the gamma oscillations representing local neural activity and ripples specific to CA1 pyramidal layer are preserved.
- Results (Figure 1): The legend mentions dotted horizontal lines from seven channels used, however, there are only four dotted lines in the actual figure.
- *Results (Section 3.6./Figure 2): The authors make quantitative claims regarding the immunohistochemistry (e.g., rows 442-443, 447-449) but do not provide quantitative data anywhere in the paper, only images taken from one sample. Quantification should be done and presented for the readers to see how much variation there were between the animals and what are the magnitudes of the seen changes.
- Results (Section 3.2./rows 247-249): “PSD plots suggest an increase in the contribution of the higher frequency bands in AD mice”. What do the authors mean by “higher frequency bands” (which bands)? Also, it is not possible to see this increase from Supp.Figs 2B-D; the authors should provide an inset zooming into the higher frequencies.
- Across the paper: why are the authors using a division operator instead of a dash when specifying bands (i.e. “10÷25 Hz” instead of “10-25 Hz”); I have never seen this notation before in this context.
- * Results (Section 3.3./Figure 4): The authors use a “Generalized Linear Model (GLM) index” to quantify phase-amplitude coupling, and compare these indices for example in Figure 4. However, there is no information anywhere in the paper about how this GLM index is calculated. The authors only refer to Supplementary Materials, but in the Supplementary Materials, they do not provide any details on how this was done, only refer to [3], which is a work of Penny et al. Penny et al have used GLM in their work to quantify phase-amplitude coupling in theta (4-8 Hz) and high gamma (76-200 Hz) bands. However, the authors here are extending it to SO and delta bands without giving any proof whether the model is applicable at all in those frequencies. Specifically, as the data acquisition system used by the authors seems to be not configured to acquire frequencies below 1 Hz (again based on Supplementary Figure 2), the phase and amplitude extracted from this data are not meaningful. Please also note the grave problems in the recording of SOs with the present techniques (see comments 4-5).
- * Results (Figure 6 and 9): In both figures, the authors report results from a very high number of parallel statistical tests. However, there are no corrections for multiple testing to assess the statistical significance of the data. With e.g., 560 parallel statistical comparisons reported in Figure 9, the authors should definitely apply some multiple testing correction before accepting any of the results as statistically significant.
- * Results (Section 3.5., rows 428-430): The authors state that “Indeed, humans carrying the PS2 mutation linked to FAD have a high incidence of silent seizures and epileptiform activity in the initial phase of the disease”, referring to the work of Jayadev et al. However, reading through Jayadev et al., this seems not to be the case: that study reports clinical seizures in PSEN2 patients, and has *zero* mentions about silent seizures or epileptiform activity. Actually, one of the references used by the authors (number 25, Lam et al) does speak about silent hippocampal seizures and epileptiform activity: in this study, Lam et al specifically disclaim that “we found *no* definite pathogenic variants in genes known to cause early onset, autosomal dominant AD (PSEN1, PSEN2, APP)“. In summary: the authors’ interpretation of the literature is in stark contrast with what can be found the papers they cite in this regard.
- *Discussion (Section 4.3., rows 526-530): The authors state that the high spiking activity/burst duration present in both young AD mice and old WT mice is “probably linked to accelerated aging” as it is independent of amyloid-beta accumulation/seeding. “Accelerated aging” means increased speed of the aging process. However, the authors only provide data at specific cross-sections of time, and at times the differences seen at an earlier timepoint seem to vanish toward a later timepoint (e.g., 3/6mo vs 12mo in Fig 8; UP-state duration and burst duration). Therefore, I do not think this data gives any foothold to make conclusions about the rate of aging.
- Section 4.4. is missing.
- Discussion (Section 4.5., rows 550-553): “[…], a finding suggestive of a direct role of astrocytes” - The data presented in this manuscript provide no evidence whatsoever about the functional role (direct or indirect) of astrocytes in the effects seen.
Author Response
Reviewer 4
Comments and Suggestions for Authors
This study is on spontaneous intracranial EEG signatures and immunohistochemical presentation of mice at a wide range of ages, and their transgenic counterparts carrying one of the dominantly inherited genes of Alzheimer’s disease (presenilin 2). While I think the experimental design is interesting, there are major issues with the presentation of the data and its interpretation. The comments below are made in a page-by-page manner, with the most important ones indicated by an asterisk (*).
Reply: The Reviewer proves to have expertise in electrophysiology and we thank him/her for the time spent reviewing our text. From his/her comments, however, there seems to be confusion regarding the data presented. From his/her reflections it appears that the Reviewer considers our acquisition system as a system for intracranial type signals (EEG), while it is a system for the acquisition of intracortical LFP signals with a multi-electrode linear probe.
For the reasons listed below (Rationale), we are reluctant to accept all the criticisms directed against the goodness of our data, acquired according to the explained methodology and following all the indications necessary to filter the input signals correctly.
Rationale
Please find below a detailed explanation of the rationale behind measurements based on LFP and EEG.
The concepts Local Field Potential (LFP) and electroencephalogram (EEG) should not be confused. Indeed, EEG signals mainly sample electrical activity recorded at the surface of the scalp by using macro-electrodes characterized by low spatial resolution. Instead, LFP signals are a measure of the electric potential recorded in the extracellular space within brain tissues, typically using micro-electrodes inserted at desired deep locations guaranteeing a fine spatial resolution investigation.
The characteristics of the LFP waveform, such as the amplitude and frequency, depend on the proportional contribution of the multiple sources and various properties of the investigated brain tissue. Specifically, the LFP is a wide-band signal containing both action potentials and other membrane potential-derived fluctuations in a small neuronal volume, thus yielding the most informative signal, e.g., for studying cortical electrogenesis. As a consequence, the larger the distance of the recording electrode from the current source, the less informative the measured LFP becomes about the events occurring at the location(s) of the source(s).
So, it is important to highlight that, although the source of neuronal signals extracted from EEG are post-synaptic extracellular currents, i.e., the same currents that contribute to spike-free LFP signals, there are several differences between invasively and non-invasively acquired signals.
First, the number and type of neurons. Indeed, since electric fields produced by neurons decay exponentially with distance, the number of neurons that have to be simultaneously active in a confined area for the fields to superimpose and produce a detectable signal is significantly smaller for LFP than EEG signals. So, this means that LFPs primarily reflect synchronized activity on a local spatial scale.
Secondly, the signal composition. Indeed, non-invasive EEG signals mainly allow analysis of low-frequency neuronal activity (i.e., <≈90 Hz), while invasive LFP signals convey information up to several kHz, since it is known that tissue acts as a low-pass filter with the subsequent undesired effect of attenuating high-frequency signals’ components.
Thirdly, the spatial distortion. Specifically, the extracellular space is composed of media with different electrophysiological properties which influence how fields spread before being detected. So, what happens is that if fields also spread in the cerebrospinal fluid, skull, and scalp, this causes even further spatial distortion before reaching the EEG electrodes.
So, overall, these limitations are intrinsic to EEG and cannot be practicably overcome.
On the contrary, there are many reasons why the use of LFP signals has become popular over the last decades. One of the most important reason is that LFPs and their different band‐limited components (known e.g. as alpha, beta and gamma bands) are invaluable for understanding different cortical functions. Indeed, cortical LFPs typically contain a very broad spectrum of oscillations of neural activity, that span a wide range of frequencies ranging from less than one Hz to one hundred Hz or more (Kayser and Konig, 2004; Lakatos et al., 2005; Buszaki, 2006; Senkowski et al., 2007), and this broad band range of activities most likely reflects contribution of several different neural processing pathways that can be, therefore, widely investigated.
R4.1 The title of this work makes a bold statement, which is not sufficiently backed up by the data. Specifically, authors interpret the EEG changes seen as “accelerated aging”. However, the current widely accepted view of aging (see e.g., The Hallmarks of Aging, López-Otin et al, Cell 2013) does not include EEG dynamics. Therefore, while there is some consistency in the interpretation that young AD mice resemble old WT mice in their EEG dynamics, the title should reflect much more precisely what was actually seen. This issue is of particular importance in the present study because the canonically accepted hallmarks of aging are unidirectionally progressive over time, whereas the EEG dynamics reported by the authors are reversed at middle age in AD mice.
Secondly, mice do not have Alzheimer’s disease (AD), and the presenilin 2 mutation is present in a small minority of all human AD cases (not important in the late-onset AD, which represents 95% of all cases, and the least common mutation of the familial early onset AD). Therefore, I do not think it is justified that the title speaks about “early stage of Alzheimer’s disease”. Rather, it should include a phrase such as “PS2APP mouse model of Alzheimer’s disease” or “PS2APP mice” or “transgenic mouse model of Alzheimer’s disease” or equivalent, like the previous papers of the authors utilizing the same model have done.
Reply: We disagree with the Reviewer’s comment on the title, also because this issue was not raised by the other Reviewers. We believe it might be simply a semantic problem. Accelerating aging occurs at the start of the disease with the rate of changes being larger at the beginning. Indeed, not all the alterations in LFP dynamics recover at middle-age but only the power ratios while other alterations were still present at this age, such as the loss of SO connectivity and reduction in SO-PAC. See also the Reply to R4.14.
We purposely did not specify the mouse model with the aim to reach a larger audience given that our observations might be of interest for researchers using other AD models as well as those studying human subjects, and help to shed light on SO.
Concerning the canonically accepted hallmarks of aging, the mentioned review is from 2013, it should be updated on the basis of more recent publications (see below). We also discussed the relevance of a longitudinal study with QEEG in classifying healthy aging subjects as Decliners and Non-Decliners, on the basis of the increase in the relative theta power (ref. 5). We have now added another reference on this issue and more can be found here on EEG biomarker linked to aging:
Predicting Age From Brain EEG Signals—A Machine Learning Approach
Al Zoubi et al. 2018. Front. Aging Neurosci., 02 July 2018 | https://doi.org/10.3389/fnagi.2018.00184
Decreased Global EEG Synchronization in Amyloid Positive Mild Cognitive Impairment and Alzheimer's Disease Patients-Relationship to APOE epsilon4.
Smailovic U, Johansson C, Koenig T, Kåreholt I, Graff C, Jelic V.
Brain Sci. 2021 Oct 16;11(10):1359. doi: 10.3390/brainsci11101359.
PMID: 34679423
Age-Related Differences in Resting-State EEG and Allocentric Spatial Working Memory Performance.
Jabès A, Klencklen G, Ruggeri P, Antonietti JP, Banta Lavenex P, Lavenex P.
Front Aging Neurosci. 2021 Nov 4;13:704362. doi: 10.3389/fnagi.2021.704362. eCollection 2021.
PMID: 34803651
R4.2 Materials and Methods/Animal Preparation and Surgery (lines 112-117): please provide the stereotaxic coordinates of the probe.
Reply: We have done it, see Supp. Materials lines 13-14.
R4.3. Materials and Methods/Signal Acquisition (lines 146): Supplementary Materials do not have any information about the quantitative evaluation of anesthesia level.
Reply: Indeed we did not do a quantitative evaluation of the level of anesthesia. We tested the absence of reflexes, as it is usually done when working with mice. Notwithstanding, as described in Materials and Methods (now in Supp. Materials in the revised version), a preprocessing step was introduced to ensure that all animals were stable during the acquisition procedure. This unbiased procedure identifies the stable heart rate and respiratory windows in which analyses were performed.
R4.4.* Results (sections 3.2, 3.3, 3.4): The authors do not report anywhere the low cutoff frequency of their EEG data acquisition system. Moreover, as the Supplementary Figure 2 shows that the PSD has its apex at 1Hz and decreases by almost two orders of magnitude from 1Hz to 0.1Hz, it is evident that the effective low cutoff frequency of the system is around 1Hz. Therefore, the data on slow oscillations (SO) in the frequency band 0.1-1 Hz are not valid.
Reply: The amplifier used for this set of experiments (Intan RHD2000 head stage) has an adjustable cutoff frequency between 0.1 and 500 Hz. These cutoff frequencies can be set through the software supplied with the acquisition system (OpenEphys GUI). The characteristics of the amplifier are described in the manufacturer's datasheet and we routinely tested its effectiveness to guarantee the reliability of acquisition and trust our data.
We accept the criticism regarding the lack of specific data on the cutoff frequencies ad we have added the upper and low cut-off frequency, with other technical specifications, of our head stage inside the text in Supplementary Materials, lines 22-23.
We reject the criticism regarding Low frequency analysis. PSD plots similar to those shown in our manuscript are also found in other studies, involving anesthesia or sleep (see blue traces in the Figure below). Moreover, the fact that we observed significant differences in that range only in the young AD cohorts is suggestive of a specific phenomenon that deserves investigation. Interestingly, middle-aged and old mice showing a much lower decrease in these frequency range (0.1-0.5 Hz) (Supp. Fig. 3C, D).
Please also note that PSD plot are in a log-log scale that amplifies the very low range.
It is worth mentioning that in humans, analyses of low frequencies by EEG recordings in sleeping subjects, have demonstrated that the loss of slow wave activity (SWA, <1 Hz, associated to NREM) predicts levels of Aβ aggregation as measured by PET, yet this is not true for delta waves (1-4 Hz) (Mander et al., 2015, doi:10.1038/nn.4035), see also refs. 51 & 52 in our work. We have now further stressed this issue, in the Discussion referring to this type of analyses, see lines 571-573.
Fernandez et al. 2016 - Highly Dynamic Spatiotemporal Organization of Low-Frequency Activities During Behavioral States in the Mouse Cerebral Cortex. - Cerebral Cortex, 2016; 1–19 doi: 10.1093/cercor/bhw311
R4.5.* Results (Figure 1): By the power law of neural oscillations, slow frequency oscillations provide the major contribution to the total power over all frequencies. Because of SO recorded in this study seem to be not valid, also the total power including these frequencies should not be used. Also, because of the disproportionate contribution of slow frequencies, the plots in Figure 1 should rather be shown for each frequency band separately.
Reply: We reject the criticism regarding the validity of our acquisitions, see general comment and above. It is worth noting that our conclusions are based on both total and relative powers as well as on power ratios, that allow comparisons between groups independently of the absolute values. In Leparulo et al. 2020 (Supp. Materials), we provided separated estimations of total power for each band at the all the investigated depths.
R4.6.* Results (Figure 1): The plot for WT mice at 16 months of age seems as if these mice have almost lost all of their EEG activity across the depth probed. Again, due to power law, this is likely due to a remarkable shift in slow oscillation dynamics. Please show the raw EEG traces at least from CA1 (1500um) and DG (2100um) depths so that the readers can see how does this remarkable difference between 12 and 16 months look in the EEG. The band-specific representation requested in comment 5 above would also make it visible whether the change seen is across all modes of neural activity, or whether the gamma oscillations representing local neural activity and ripples specific to CA1 pyramidal layer are preserved.
Reply: These are not EEG recordings, so with depth, being a linear probe, we do not lose activity. We include here a figure showing raw LFP traces, at required depths, in 12- and 16-month-old WT mice showing how activity is preserved.
R4.7.* Results (Figure 1): The legend mentions dotted horizontal lines from seven channels used, however, there are only four dotted lines in the actual figure.
Reply: The problem arises from the conversion of the image from the original format to the one used for the pdf file, which causes some features to lose definition. Unfortunately, also it occurs even when using less compressed versions of the image. We are sorry for this problem.
R4.8. * Results (Section 3.6./Figure 2): The authors make quantitative claims regarding the immunohistochemistry (e.g., rows 442-443, 447-449) but do not provide quantitative data anywhere in the paper, only images taken from one sample. Quantification should be done and presented for the readers to see how much variation there were between the animals and what are the magnitudes of the seen changes.
Reply: We have now quantified amyloidosis and gliosis as estimated by 4G8 and GFAP staining respectively in Supp. Fig. 2. We have not done it in the original manuscript because this is not a novel AD mouse model. In fact, regarding this aspect, it has already been characterized (Richards et al. ’03; Ozmen et al ’09; Fontana et al. ’17).
R4.9. Results (Section 3.2./rows 247-249): “PSD plots suggest an increase in the contribution of the higher frequency bands in AD mice”. What do the authors mean by “higher frequency bands” (which bands)? Also, it is not possible to see this increase from Supp.Figs 2B-D; the authors should provide an inset zooming into the higher frequencies.
Reply: The higher frequency bands are defined with respect to SO and delta waves (> 4.7 Hz), i.e. the theta, beta, FG and epsilon as schematically shown in Supp. Fig. 3A and mentioned in the text (lines 200-204). We have now included a zooming into the PSD function at higher frequencies (90-190 Hz), as log-semilog plot, see Supp. Fig. 3E, F and line 207.
R4.10. Across the paper: why are the authors using a division operator instead of a dash when specifying bands (i.e. “10-25 Hz” instead of “10÷25 Hz”); I have never seen this notation before in this context.
Reply: In the revised version we opted for the notation suggested by the Reviewer, as also used in our previous works.
R4.11. * Results (Section 3.3./Figure 4): The authors use a “Generalized Linear Model (GLM) index” to quantify phase-amplitude coupling, and compare these indices for example in Figure 4. However, there is no information anywhere in the paper about how this GLM index is calculated. The authors only refer to Supplementary Materials, but in the Supplementary Materials, they do not provide any details on how this was done, only refer to [3], which is a work of Penny et al. Penny et al have used GLM in their work to quantify phase-amplitude coupling in theta (4-8 Hz) and high gamma (76-200 Hz) bands. However, the authors here are extending it to SO and delta bands without giving any proof whether the model is applicable at all in those frequencies. Specifically, as the data acquisition system used by the authors seems to be not configured to acquire frequencies below 1 Hz (again based on Supplementary Figure 2), the phase and amplitude extracted from this data are not meaningful. Please also note the grave problems in the recording of SOs with the present techniques (see comments 4-5).
Reply: We are sorry but the GLM was introduced in Supp. Materials (Rubega et al 2016, now ref. 5). In the past, our group has thoroughly investigated this problem demonstrating that the General Linear Model (GLM) is a better and safer estimator of PAC than the modulation index (MI), being less affected by signal noise and the length of the signal chunk. We used this approach in our works (Fontana et al. '17, Leparulo et al. '20). This information was already present in the old Supp. Materials (now at line 72 of the new Supp. Materials: Finally, PAC index was computed by means of the general linear model (GLM) as described in [4]. The GLM method was chosen owing to its sensitivity [5]. Basically, the amplitude of the faster oscillation is modelled by multiple regression and the index is the proportion of variance explained by the model.
R4.12. * Results (Figure 6 and 9): In both figures, the authors report results from a very high number of parallel statistical tests. However, there are no corrections for multiple testing to assess the statistical significance of the data. With e.g., 560 parallel statistical comparisons reported in Figure 9, the authors should definitely apply some multiple testing correction before accepting any of the results as statistically significant.
Reply: We did not apply multiple comparison for two reasons: i) we were exclusively interested on the comparison of each cohort versus young WT mice; ii) we decided to avoid errors of type II, and thus discard too much data, which derive from in vivo experiments, where animals, especially the old ones, are very precious. Therefore we avoided stringency and opted for planned comparison rather than every possible comparison. Indeed, statisticians recommend not doing any formal corrections for multiple comparisons when the study focuses on only a few scientifically sensible comparisons.
R4.13. * Results (Section 3.5., rows 428-430): The authors state that “Indeed, humans carrying the PS2 mutation linked to FAD have a high incidence of silent seizures and epileptiform activity in the initial phase of the disease”, referring to the work of Jayadev et al. However, reading through Jayadev et al., this seems not to be the case: that study reports clinical seizures in PSEN2 patients, and has *zero* mentions about silent seizures or epileptiform activity. Actually, one of the references used by the authors (number 25, Lam et al) does speak about silent hippocampal seizures and epileptiform activity: in this study, Lam et al specifically disclaim that “we found *no* definite pathogenic variants in genes known to cause early onset, autosomal dominant AD (PSEN1, PSEN2, APP)“. In summary: the authors’ interpretation of the literature is in stark contrast with what can be found the papers they cite in this regard.
Reply: Yes, the Reviewer is right, Lam’s work only deals with a sporadic AD patient. The fact that PS2 mutations are very rare, does not mean that they are not informative. We have learned a lot on the regulation of Ca2+ homeostasis, ER-mitochondria coupling and metabolic/autophagic dysfunctions by studying PS2 mutations in vitro and in situ as well as PS2KO mice. It is worth mentioning that also the endogenous PS2 has a direct role on Ca2+ signaling, similar to those exerted by mutant PS2, upon overexpression (see Pizzo et al. 2020 for a review). The issue of the amount of endogenous PS2, as a possible player also in SAD was already present in the Discussion (now at line 505):”it has been demonstrated that, in neurons from AD patients, early AD is accompanied by an increase in the expression level of PSEN2 mRNA and PS2 protein, through the loss of the transcriptional regulator REST [80]”. We have now inserted also a new sentence regarding mice: …”Likewise, C57Bl/6J mice also show increased PS2 levels with aging [81]”, a phenomenon that is likely even more relevant given the downregulation of PS1 with aging.
R4.14. *Discussion (Section 4.3., rows 526-530): The authors state that the high spiking activity/burst duration present in both young AD mice and old WT mice is “probably linked to accelerated aging” as it is independent of amyloid-beta accumulation/seeding. “Accelerated aging” means increased speed of the aging process. However, the authors only provide data at specific cross-sections of time, and at times the differences seen at an earlier timepoint seem to vanish toward a later timepoint (e.g., 3/6mo vs 12mo in Fig 8; UP-state duration and burst duration). Therefore, I do not think this data gives any foothold to make conclusions about the rate of aging.
Reply: We simply observed that the MFR and burst duration, which are significantly increased in PS2.30H mice at 3 months and in B6.152H mice at 6 months, are similarly changed in WT at 16 months. We consider these as features of anticipated aging, i.e. of something that takes place in old WT mice, yet can partially compensate ad older age. Indeed, adaptation to aging signs it has been described at many levels. See for instance:
- The Aging Brain: Functional Adaptation Across Adulthood – 2018 Edited by Gregory R. Samanez-Larkin.
- Zia et al. 2021 Molecular and cellular pathways contributing to brain aging. Behav Brain Funct 17, 6 (2021). https://doi.org/10.1186/s12993-021-00179-9
- Lee et al. 2015, Adaptation of Brain Functional and Structural Networks in Aging https://doi.org/10.1371/journal.pone.0123462
R4.15. Section 4.4. is missing.
Reply: Ok, thanks.
R4.16. Discussion (Section 4.5., rows 550-553): “[…], a finding suggestive of a direct role of astrocytes” - The data presented in this manuscript provide no evidence whatsoever about the functional role (direct or indirect) of astrocytes in the effects seen.
Reply: In our opinion, the discussion chapter should contain hypotheses based on the obtained data but also on related literature. By considering the following aspects: i) the role of PS2 on Ca2+ dysregulation at many level, including astrocytes; ii) the fact that these tg models express the mutant PS2 in astrocytes, iii) the relevance of Ca2+ signaling on astrocyte excitability, which is not questionable together with the findings by Poskanzer & coworkers, refs. 84, 85, iv) the concept of tripartite synapse, and the role of gliotrasmission on slow oscillations (Fellin et al. 2009, https://doi.org/10.1073/pnas.0906419106), we believe that alterations of Ca2+ homeostasis as well metabolic imbalances, similar to those caused by mutant PS2, have a high probability to be involved in UP-DOWN states. Notwithstanding, other hypotheses have been advanced to explain our findings.

Round 2
Reviewer 3 Report
The authors did not significantly improve the manuscript according to my review.
Reviewer 4 Report
General Comment:
I was fully aware during the whole review process that the Authors indeed used an Open Ephys – Intan – Atlas Neuroscientific system to record intraparenchymal/intracortical local field potentials (LFPs).
I apologize for the semantic confusion stemming from the terms “intracranial EEG” vs “intracortical LFP”, which I have seen being used interchangeably; however I agree with the Authors that “LFP” is a better and more accurate term to use here. Anyhow, I must emphasize that none of my comments were made under assumptions that the Authors would have used any other recording methodology than the one mentioned above.
I am also convinced on the usefulness of LFP recordings in studying brain activity/function experimentally: these recordings indeed give valuable information with high spatiotemporal resolution, which cannot be acquired with clinical EEG – I agree with the Authors here, but my critical comments were not pointing towards this matter.
R.4.4. The most important critic I had regarding this study is its analysis and interpretation of data on slow oscillations (SO, defined as 0.1-1.7Hz). The Authors have now provided the cutoff frequencies used, which they state as 0.1Hz for low cutoff and 500Hz for high cutoff. However, based on the data of the manuscript, and the reference they give in their response, I continue to have concerns about their slow oscillation LFP data, and therefore their conclusions based on the SO phenomena in this paper. More specifically:
1. In their response, the Authors refer to Fernandez et al. 2016 - Highly Dynamic Spatiotemporal Organization of Low-Frequency Activities During Behavioral States in the Mouse Cerebral Cortex. - Cerebral Cortex, 2016; 1–19 doi: 10.1093/cercor/bhw311
The graphs presented in Fernandez et al. Figure 3 are not PSDs (power spectral densities), because their unit is uV/Hz [amplitude per frequency]: this is an amplitude-frequency spectrum instead of a power-frequency spectrum (which has the unit of mV^2/Hz [amplitude squared per frequency]).
As power is the square of amplitude, a hundred-fold decrease in PSD (which is seen between 1Hz and 0.1Hz in the mean PSD, Supplementary Fig 3A) corresponds to a ten-fold decrease (90% drop) in amplitude spectrum. However, in Fernandez et al. Figure 3A, the mean amplitude in CA1 under NREM goes down from approximately 33 uV/Hz at 1Hz to 25 uV/Hz at 0.1Hz, which is ~25% drop. Therefore, there is a considerably larger attenuation of slow oscillations in the currently reviewed manuscript compared to Fernandez et al.
2. The PSD presented in Supplementary Fig 3A decreases very smoothly without a single bump from 1Hz to 0.1Hz, which is not seen in frequencies higher than 1Hz.
The Authors say in Supplementary Methods that “baseline drift was removed from all signals using a median estimation method”. Maybe this explains some of my concerns above, as according to the 1/f noise law, the slow frequencies have the highest noise. However, as the slowest oscillations are most widespread spatially in the brain, there is a risk that baseline drift removal actually partly removes also meaningful slow oscillatory activity. Could the Authors explain how exactly does their median estimation method work, and provide a PSD before the baseline drift removal?
Other comments:
R4.6. See above regarding the EEG vs LFP terminological confusion. With “losing activity”, I mean that in Fig 1A, the mean total power of 16 mo mice is near zero across the depth profile. However, I do think that the figure of raw LFP traces that the Authors have provided me considerably clarifies the situation, as it shows that the activity in 16mo group looks fine but is just very low amplitude compared to the younger age groups. I am sure that also the other readers of this publication would greatly value the raw trace examples that the Authors have shown me, so I suggest they would be presented for instance in the Supplement.
R4.11. I checked both Fontana et al. 2017 and Leparulo et al 2020, which both have the lines that the Authors have italicized in their response, ultimately referring back to Rubega et al. 2016. However, that paper is not accessible to me even via institutional access. Could the authors provide briefly the equation that the GLM uses to model the oscillations? I am sure many readers would also consider this information interesting, in case GLM is a preferable method over MI for estimating PAC as the Authors suggest.
R4.12. The Authors reply that they have avoided statistical stringency while analyzing their results to “avoid errors of type II”, which in plain English means that they want to avoid false negative results. While doing this by not correcting for 560 parallel comparisons, they considerably increase their probability of making errors of type I, i.e., false positive results.
The Authors do not clarify this choice in the manuscript, and do not advice to take any of the results with caution, but instead make straightforward conclusions from their results: “this study allows us to conclude that specific features of the aging process, […], occur in the early stages of AD.” I think the Authors should clarify in the manuscript that because of these statistical choices, the results in Figs 6, 9 and Supplementary Fig 9 should be taken as suggestive/hypothesis-generating and must be studied further with more stringent statistical criteria.
R4.13. The authors have not addressed my comment. The comment was not focusing on Lam’s work, but the statement in the Authors’ work saying “Indeed, humans carrying the PS2 mutation linked to FAD have a high incidence of silent seizures and epileptiform activity in the initial phase of the disease [71]”. This is not true. The work by Jayadev et al which is referred does not back up the statement by the Authors. Please remove this claim or present reference which actually shows the link between humans carrying the PS2 mutation and a “high incidence of silent seizures and epileptiform activity in the initial phase of the disease”.
For other parts, I consider the Authors have sufficiently addressed my comments.